

**Improved Atmospheric Characterization through Fused Mobile**
**Airborne & Surface *In Situ* Surveys: Methane Emissions**
**Quantification from a Producing Oil Field**
Ira Leifer[1], Christopher Melton[1], Marc L. Fischer[2], Matthew Fladeland[3], Jason Frash[1], Warren Gore[3],
Laura Iraci[3], Josette Marrero[3], Ju-Mee Ryoo[3], Tomoaki Tanaka[3], Emma Yates[3]
[1]Bubbleology Research International, Solvang, CA 93463, ira.leifer@bubbleology.com
[2]Lawrence Berkeley National Laboratory, 1 Cyclotron Road, Berkeley CA 94720.
[3]NASA Ames Research Center, Moffett Field, CA, 94035
**Correspondence to:** Ira Leifer (Ira.Leifer@bubbleology.com
**Abstract.** Methane ($CH_4$) inventory uncertainties are large, requiring robust emission derivation
approaches. We report on a fused airborne/surface data collection approach to derive emissions from an
active oil field near Bakersfield, central California. The approach characterizes the atmosphere from the
surface to above the planetary boundary layer (PBL) and combines downwind trace gas concentration
anomaly (plume) above background with normal winds to derive flux. This approach does not require a
well-mixed PBL, allows explicit, data based, uncertainty evaluation, and was applied to complex
topography and wind flows.
*In situ* airborne (collected by AJAX – the Alpha Jet Atmospheric eXperiment) and mobile surface
(collected by AMOG – the AutoMObile trace Gas – Surveyor) data were collected on 19 August 2015 to
assess source strength. Data included an AMOG and AJAX intercomparison transect profiling from the
San Joaquin Valley (SJV) floor into the Sierra Nevada Mountains (0.1-2.2 km altitude), validating a novel
surface approach for atmospheric profiling by leveraging topography. The profile intercomparison found
good agreement in multiple parameters for the overlapping altitude range from 500 to 1500 m, for the
upper 5% of surface winds, which accounts for wind-impeding structures, i.e., terrain, trees, buildings,
etc. Annualized emissions from the active oil fields were 31.3±16 Gg methane and 2.4±1.2 Tg carbon
dioxide. Data showed the PBL was not well-mixed at distances of 10-20 km downwind, highlighting the
importance of the experimental design.



**1.    Introduction**
**1.1. Methane Trends and Uncertainty**
On decadal timescales, methane ($CH_4$), affects the atmospheric radiative balance more strongly than
carbon dioxide ($CO_2$) (IPCC, 2007, Fig. 2.21). Since pre-industrial times, $CH_4$ emissions have risen by a
factor of 2.5 (Khalil and Rasmussen, 1995), while estimates of its lifetime has decreased and now is
estimated at ~8.5 years (Sonnemann and Grygalashvyly, 2014). Atmospheric $CH_4$ growth almost ceased
between 1999 and 2006, but has resumed since 2007 (Nisbet et al., 2015). Several processes are proposed
to underlie this trend (Ghosh et al., 2015; John et al., 2012; Nisbet et al., 2015); however, high uncertainty
in emission inventories (IPCC, 2013) complicates interpretation of the underlying mechanism(s).

The dominant $CH_4$ loss arises from reaction with hydroxyl (OH), whose concentration has been
increasing in recent decades (John et al., 2012), causing a decrease in the estimated $CH_4$ lifetime of 0.5%
$yr^{-1}$ (Karlsdóttir and Isaksen, 2000). Overall, the estimate of the $CH_4$ lifetime has decreased by ~40% from
an estimated 12 years in 2007 (IPCC, 2007). The recent discovery of a new significant $CH_4$ loss
mechanism, terrestrial uptake (Fernandez-Cortes et al., 2015), illustrates the need to understand loss
mechanisms better.

Large $CH_4$ budget uncertainties remain for many sources (IPCC, 2013), with greater uncertainty in future
trends from global warming feedback (Rigby et al., 2008) and increasing anthropogenic activities
(Kirschke et al., 2013; Wunch et al., 2009). Emphasizing these uncertainties are recent studies that
suggest underestimation by a factor of 1.5 in the important anthropogenic $CH_4$ source, Fossil Fuel
Industrial (FFI) emissions (Brandt et al., 2014). Tellingly, this discrepancy only was noted recently
(Miller et al., 2013), in part because the US $CH_4$ monitoring network is too sparse to constrain emissions
at "regional to national scales" (Dlugokencky et al., 2013). FFI emissions are the most (Brandt et al.,
2014; EPA, 2017) anthropogenic contributor to the global $CH_4$ budget. Whereas EPA inventory values
and Bruhwiler et al. (2017) suggest no significant trends in the north American emissions over the last
decade, satellite and surface observations suggest a 30% increase in US $CH_4$ emissions (Turner et al.,
2016). However, Turner et al. (2016) could not ascribe a specific source. These uncertainties strongly
argue for the need for new, robust methodologies for flux derivation.




**1.2. Methane Flux Estimation**
Various approaches have been developed to derive surface emissions from $CH_4$ concentration
measurements including direct assessment (Peischl et al., 2015; White et al., 1976), data-driven mass
balance, e.g., Karion et al. (2013), tracer-tracer ratio (LaFranchi et al., 2013), and assimilation inverse
models, e.g. Jeong et al. (2013); Jeong et al. (2012). Challenges for the latter approach include the needs
for accurate meteorological transport models and good *a priori* emission distributions (Miller et al.,
2013). Miller et al. (2013) concluded that bottom-up inventories (EPA, 2013; European Commission,
2010) significantly underestimate husbandry and FFI emissions. To apportion $CH_4$ to FFI versus
biological sources, the tracer-tracer approach has been applied using ethane, whose emission ratio to $CH_4$
requires tight constraint (Peischl et al., 2013; Simpson et al., 2012; Wennberg et al., 2012). In practice,
this emission ratio is an *a priori* assumption in the approach.

Direct assessment approaches have advantages over inversion approaches. Direct approaches allow
explicit uncertainty evaluation and do not require an *a priori* emission spatial distribution, which may be
unknown. Direct approaches also do not require the ability to model atmospheric transport accurately
across the study region. In areas of complex topography or highly variable winds, this transport can
challenge assimilation approaches, which also are challenged in areas with poorly characterized (or
unknown) or highly variable sources, particularly if the measurement network is sparse. For direct
assessment approaches, data collection should be rapid if winds and/or emissions are variable, and at
adequate data density to characterize fine-scale structure.
**1.3. Study Motivation**
Herein we report on a novel application of fused airborne and surface *in situ* data to directly estimate $CH_4$
emissions. Specifically, on 19 August 2015, NASA's Alpha Jet Atmospheric eXperiment (AJAX)
collected 1164 km of airborne data while AMOG (AutoMObile greenhouse Gas) Surveyor collected 1074
km of contemporaneous mobile surface data. Both measure carbon dioxide ($CO_2$), $CH_4$, water vapor
($H_2O$), and ozone ($O_3$), as well as winds, pressure, relative humidity (*RH*), and temperature (*T*). These
surface and airborne datasets were collected in a downwind curtain or plane oriented approximately
orthogonal to the winds, to characterize the full planetary boundary layer (PBL) from surface to above the
PBL. Additionally, the survey route was designed to include an ascent to ~2.2 km above sea level to
include surface PBL characterization. Data fusion between platforms was validated by a vertical profile
intercomparison for 0.5 to 1.5 km altitude by AMOG SURVEYOR leveraging topographic relief.
Leveraging topographic relief –mountainous terrain affects about half the earth's population and about
half the earth's land surface (Meyers and Steenburgh, 2013) – allows a surface platform to collect
atmospheric profile data and is a useful research tool in the absence of airborne resources.
**1.4 The South San Joaquin Valley, California**
Most of California oil production lies in the San Joaquin Valley (SJV), as does most of California
agriculture, including many intensive dairies (Gentner et al., 2014), and the major north-south
transportation artery. For this study, data were collected for the Kern River oil fields (Kern Front oil field,
Kern River oil field and the Poso Creek oil field, referred to herein as the Kern Fields), located adjacent to
northwest Bakersfield (**Fig. 1A**). These adjacent oil fields create a strong $CH_4$ source that largely is
isolated from confounding plumes from other SJV sources. This area includes complex wind flow
patterns across and around the "toe" of Sierra Nevada Mountain foothills, which extend into the Kern
Front and Kern River oil fields. Here, topographic steering ensures predictable prevailing northwesterly
winds blow across the Kern Fields.
Strong orographic forcing also arises from tall bluffs (~100 m) on the Kern River Valley's south bank,
which also separates the Kern River oil field from the urban city of Bakersfield (pop. 364,000 in 2013).
The fine-scale wind structure that results from orographic forcing on transport dictated an anomaly
approach for flux derivation, as did the presence of strong $CH_4$ structures (plumes) in the valley's lowest
air. In the anomaly approach, transects must extend beyond a reasonably well-defined plume.

Topography (i.e., mountain ranges) plays a locally dominant role in overall southern California air flows
where upper level winds locally force the lower level flows that transport pollutants (Bao et al., 2008).
The SJV is delimited on the east by the Sierra Nevada Mountains and on the west by the Transverse
Coastal Mountain Range (**Fig. 1A**). Transport between the SJV and adjacent air basins is poor due to
California's mountain ranges. The SJV features weak surface winds (Bao et al., 2008) with the worst air
quality in the United States occurring in the cities of Bakersfield and Delano (American Lung
Association, 2016) in the SJV.

Pacific Ocean air primarily enters the SJV through the San Francisco Bay area and the Carquinez Strait,
where it splits north into the Sacramento Valley and south into the SJV (Zhong et al., 2004). This flow
extends up to ~1 km altitude. These winds are near orthogonal to the 600-km long central valley of
California  - i.e., cross-slope. South of Bakersfield, winds shift to from the west due to mountains that
guide SJV air out into the Mojave Desert, where it affects air quality for up to hundreds of kilometers
distance (VanCuren, 2015). Although the Tehachapi Pass is the main exit pathway of SJV air, other
passes also transport air into the Mojave Desert. These flows are augmented by high inland temperatures





relative to the Pacific Ocean, which creates a horizontal pressure gradient that drives local upslope flows
during the day and returning downslope nocturnal flows (Zhong et al., 2004). The pressure gradient is
maximal around sunset, although winds peak ~4 hours later, shortly before midnight. This pressure
gradient is controlled by the semi-permanent Pacific high, situated offshore central California, which
diverts storms far to the north during summer. This pressure feature drives prevailing west-southwesterly
winds at the regional scale in the California south coast air basins (Boucouvala and Bornstein, 2003).
**2.  Methodology**
**2.1.  Experimental design**
Data were collected as part of the *GOSAT-COMEX Experiment* (Greenhouse gases Observing SATellite -
$CO_2$ and Methane Experiment - GCE) Campaign. GCE was developed to characterize emissions on
spatial scales from decameter (*in situ* surface, imaging spectroscopy) to kilometer (*in situ* airborne) to
deca-kilometer (satellite) in an area of complex topography. GCE design combined *in situ* mobile surface
and airborne data with GOSAT satellite data. *In situ* data serve to assess the satellite pixel / plume
overlap. Key GCE requirements are relatively steady, strong, isolated emissions and predictable and
steady winds. Prevailing study area winds are from the west-northwest, veering to westerly winds to the
southeast of Bakersfield (**Fig. 1**). Prevailing wind directions are highly reliable due to topographic
control.

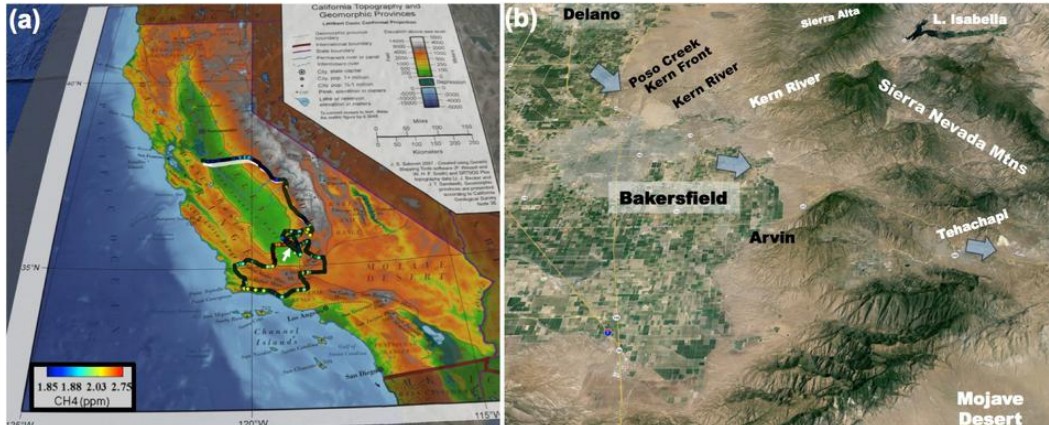


**Figure 1.** (a) Full surface and airborne data for 19 Aug. 2015 mapped over California topography. White
arrow shows Bakersfield. Data key on panel. (b) Study area map showing direction of prevailing winds
and nearby mountain topography (Google Earth, 2016). See Supp. Fig. S1 for a high-altitude (20-km)
photo of the entire study area and surrounding terrain.





GCE developed from the COMEX Campaign (Krautwurst et al., 2016), which combined *in situ* airborne
and surface observations with both imaging and non-imaging spectroscopy to explore synergies for GHG
emission estimation (Thompson et al., 2015). COMEX focused on southern California $CH_4$ sources
including husbandry, landfills, natural geology, and petroleum hydrocarbon refining and production.

GCE combines airborne and surface data collected at dramatically different speeds. AJAX collects data at
~500 km hr$^{-1}$, capturing a snapshot of atmospheric winds and plume structure. Surface GCE data are
collected quasi-Lagrangian, starting northwest (upwind) and proceeding southeast and then east
(downwind). This enables useful data collection even when a $CH_4$ plume drifts into the study area after
the upwind survey – data collection proceeds downwind faster than advection. The surface route was
designed carefully to traverse all targeted GOSAT pixels using rarely used (low traffic) surface roads and
requires ~100 minutes.

Airborne and surface surveys are timed so that the downwind data plane (Krings et al., 2011) is surveyed
concurrent with the satellite overpass. Data planes extend from the surface (AMOG) to above the PBL
(AJAX), reducing uncertainty by providing a more complete atmospheric characterization including
below where airplanes are permitted to fly (~500 m in an urban area). AJAX and AMOG profile data are
fused to impose vertical structure during interpolation. Surface and airborne datasets are interpolated and
fused to derive the flux passing through the data curtain (Sect. 2.5).

CGE first incorporates an upwind transit from Delano (100 m) on the SJV floor to Sierra Alta (1800 m)
and higher to confirm that stranded $CH_4$ clouds (plumes disconnected from a source) do not threaten to
impact the study area during the experiment, otherwise the survey is aborted. A key mission abort
criterion is wind compliance. Specifically, winds must not be too light or variable, must flush nocturnal
accumulations before the GOSAT overpass, and must be prevailing. The upwind transit provides vertical
profile information including PBL height and vertical structure.
**2.2. AutoMObile trace Gas (AMOG) Surveyor**
Mobile atmospheric surface measurements have been conducted for many years using a customized van
(Lamb et al., 1995) or a recreational vehicle (Farrell et al., 2013; Leifer et al., 2013). Recently, the
development of cavity enhanced absorption spectroscopy (CEAS) analyzers has opened the way for rapid
and highly accurate trace gas measurements (Leen et al., 2013) without the need for compressed gases as
in gas chromatography (Farrell et al., 2013). This allows for smaller vehicle survey platforms at lower
logistical overhead (Leifer et al., 2014; McKain et al., 2015; Pétron et al., 2012; Yacovitch et al., 2015). A





competing sensor technology that has been used in mobile survey data collection is open path
spectroscopy (Sun et al., 2014). Older technology using fluorescence also can be incorporated onto
mobile survey platforms, for example, to measure ozone, $O_3$.

Mobile surface data were collected by the AMOG Surveyor (Leifer et al., 2014) (see Supp. Sect. S2.1 for
additional details), a modified commuter car. AMOG Surveyor provides mobile high-speed, high-spatial
resolution observations of meteorology (winds, temperature, pressure), trace gases (greenhouse and
others), and remote sensing parameters. AMOG uses a range of trace gas analyzers and careful design
with respect to wind flow around the vehicle to characterize strong spatial heterogeneity at up to highway
speeds.

Two-dimensional winds are measured by a sonic anemometer (VMT700, Vaisala) mounted 1.4 m above
the roof, above vehicle flow streamlines for slow to highway speeds. Air is drawn down two sample lines
from 5 and 3 m above ground by a high-flow vacuum pump (GVB30, Edwards Vacuum) that feeds
several gas analyzers. The greenhouse gases, $CO_2$, $CH_4$, and $H_2O$, are measured at up to 10 Hz by an
analyzer that uses Integrated Cavity Offaxis Spectrometer-Cavity Enhanced Absorption Spectroscopy
(ICOS-CEAS, 911-0010, Los Gatos Research, Inc.). A fluorescence analyzer measured $O_3$ at 0.25 Hz
(49C, ThermoFischer Scientific, MA). AMOG Surveyor's full trace gas suite (carbonyl sulfide, carbon
monoxide, nitric oxide, nitrogen dioxide, hydrogen sulfide, sulfur dioxide, total sulfur, ammonia) was not
deployed on 19 Aug. 2015.

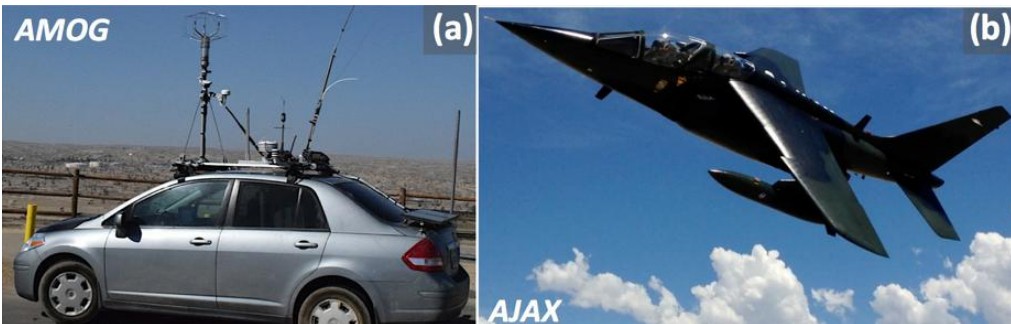


**Figure 2.** Study platforms. (a) AutoMObile trace Gas (AMOG) Surveyor, Kern River oil field in
background. Photo courtesy Ira Leifer. (b) The Alpha Jet Atmospheric eXperiment (AJAX) aircraft, photo
courtesy Akihiko Kuze, JAXA. See Supplemental Material Section 1 for further details.
Relevant recent AMOG Surveyor improvements since Leifer et al. (2014) include a high speed
thermocouple (50416-T, Cooper-Atkins) and a high accuracy (0.2 hPa) pressure sensor (61320V RM



Young Co.). Both are mounted in a roof passive radiation shield (7710, Davis Instruments) to largely
eliminate dynamic pressure effects from the airflow. Position information is critical to accurate wind
measurements and is provided by redundant (two) Global Navigation Satellite Systems (19X HVS,
Garmin) that use the GLONASS, GPS, Galileo, and QZSS satellites at 10 Hz (WGS84). AMOG
analyzers and sensor data are logged asynchronously on a single computer. Custom software integrates
the data streams and provides real-time visualization of multiple parameters in the Google Earth
environment.

### 213    2.3. Alpha Jet Atmospheric eXperiment (AJAX)

AJAX (Fig. 2b) collected airborne *in situ* measurements of $CO_2$, $CH_4$, $H_2O$ by cavity ring down
spectroscopy (G2301-m, Picarro Inc.), $O_3$, (Model 205, 2B Technologies Inc.), and meteorological
parameters including 3D winds (Meteorological Measurement System). The greenhouse gas analyzer is
calibrated using NOAA whole-air standards; calibrations are performed before and/or after each flight
with the calibration factor closest to the day of flight being applied to each raw $CO_2$ and $CH_4$
measurement. Further corrections include applying water vapor corrections provided by Chen et al.
(2010) to calculate $CO_2$ and $CH_4$ dry mixing ratios. Data also are filtered for quality control for deviations
in instrument cavity pressure, to improve inflight precision.

The overall $CH_4$ measurement uncertainty is typically <2.2 ppb, including contributions from accuracy of
the standard, precision (1- $\sigma$ over 6 min), calibration repeatability, inflight variance due to cavity pressure
fluctuations, and uncertainty due to water corrections and pressure dependence (based on environmental
chamber studies). See Hamill et al. (2015); Tanaka et al. (2016), and Yates et al. (2013) for further
aircraft and instrumentation details, and Supp. Sect. S2.2.

### 228    2.4. Background estimation and data fusion

The flux ($Q(x, z)$) with respect to lateral transect distance ($x$) and altitude ($z$) through the $x$, $z$ plane is the
product of the normal winds ($U_n(x, z)$) and the plume concentration anomaly ($C'(x, z)$). Interpolation of $C'$
and $U_N$ is linear within the PBL and is assumed uniform above the PBL.

To calculate $Q(x, z)$ requires $C'$ relative to background ($C_B(x, z)$), which is derived from evaluating
$C_B(x < x_{max}/2, z)$ and $C_B(x > x_{max}/2, z)$, denoted $C_{BL}(z)$ and $C_{BR}(z)$, respectively, where $x_{max}$ is the lateral extent
of the data curtain. Then, $C_B(x,z)$ is derived from a linear polynomial fit of $C_{BL}(z)$ and $C_{BR}(z)$.



Both $C_{BL}(z)$ and $C_{BR}(z)$ are derived from the left and right probability density functions ($\Phi_L(C(x<x_{max}/2, z))$
and ($\Phi_R(C(x>x_{max}/2,z))$), respectively, for each flight transect level. Specifically, for $\Phi_L$ and $\Phi_R$, Gaussian
functions are fit to the distributions for the plume distribution ($\Phi_P$) and the background distribution ($\Phi_B$).
In practice, $\Phi_B$ is well described by a single Gaussian, while $\Phi_P$ is best described by multiple Gaussian
functions. Then, $C_{BL}(z)$ and $C_{BR}(z)$ are defined such that,
$\int \Phi_{BL}(C_{BL}(z)) = 0$ and $\int \Phi_{BR}(C_{BR}(z)) = 0.$        (1)
where $\Phi_{BL}$ and $\Phi_{BR}$ are the background $\Phi_B$ for the left and right halves of the data plane, respectively.
Concentration is not a conserved value, thus $C'$ is converted into mass ($N'$) by the ideal gas law for spatial
integration to derive the total emissions ($E$), which is the integration of the flux through the plane, $Q$,
$$E = \int_{x1}^{x2} \int_0^{z=PBL} Q(x,z) \, dz \, dx$$        (2)
**2.5. Uncertainty evaluation for emission calculation**
A flux estimate requires two types of assumptions with respect to the flux calculation: representativeness
and appropriateness. Specifically, background concentration profiles may be incorrect, while winds,
which are measured accurately, could be un-representative, as could concentrations due to temporal
variability over the period needed to make the measurements. Monte Carlo simulations based on observed
data variability were run to assess uncertainty. Instrumental uncertainty is far less than spatial and
temporal variability and hence appropriateness is the dominant source of uncertainty.

Monte Carlo simulations were based on 1 standard deviation in $U_N(z)$ around the mean for each flight
transect altitude level. Gaussian distributions were created with half widths of 4 seconds and randomly
sampled to populate $U_N(x,z)$, which then was interpolated vertically for the flux calculation. Other
variables were allowed to vary in a similar manner and sampled by the Monte Carlo simulations. Monte
Carlo simulations addressed uncertainties in $C_B(z)$ based on the data variability at the edges of the data
plane. This addresses uncertainty in precisely from where the inflowing air is arriving, which alters the
background concentration in the flux calculation. One million Monte Carlo simulations were run for a
flux uncertainty calculation.





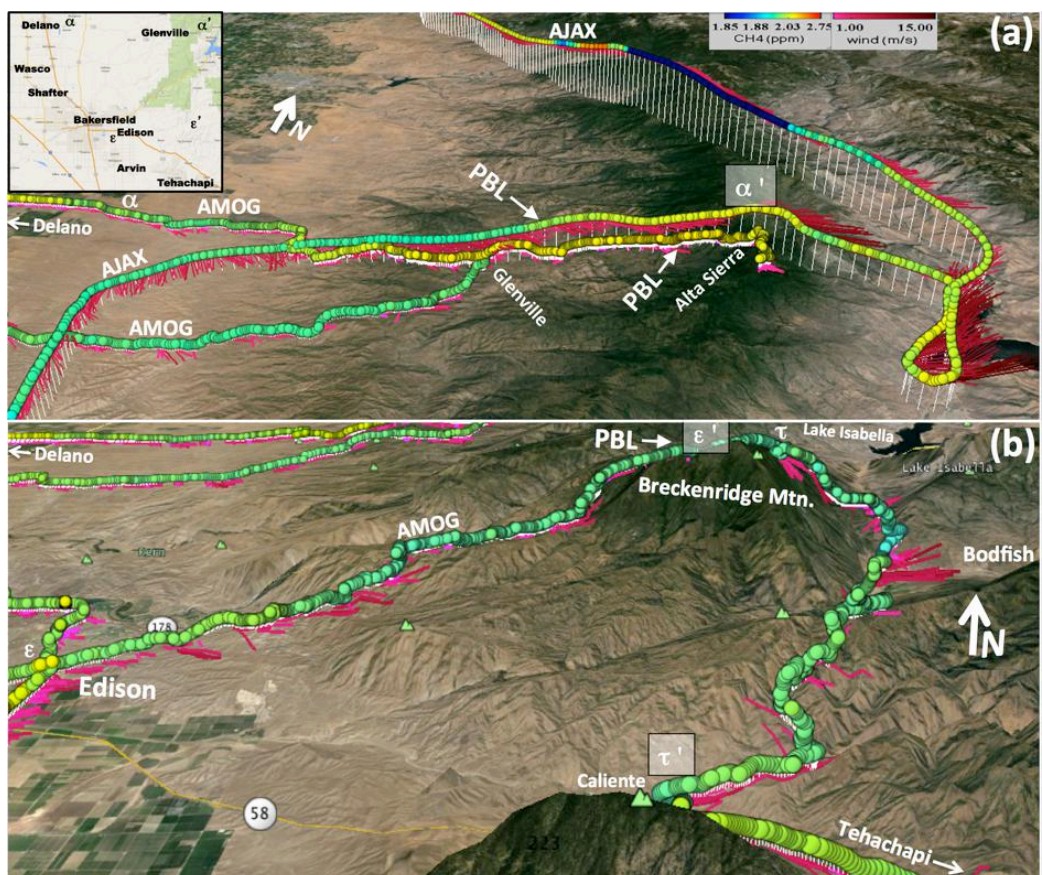


**Figure 3.** (a) Pre-survey, upwind AMOG surface and AJAX airborne methane (CH₄) and winds for vertical profile on the Delano – Alta Sierra transect (α–α'). Inset shows area map. (b) Post survey, downwind AMOG surface profile ascent Edison-Breckenridge (ε–ε') and descent Breckenridge-Bodfish-Caliente (τ–τ'). Upwind profile visible top left. Planetary boundary layer (PBL) identified.

## 3. Results

### 3.1. Profile data

Four vertical profiles (surface and airborne) were collected to understand PBL evolution during the survey (2 hrs.) and across the survey domain. AMOG and AJAX collected pre-survey intercomparison vertical profiles ~30 km north of the Kern Fields between the small town of Delano on the SJV floor (100 m) up to Shirley Meadows (2100 m) on a ridge of the Greenhorn Mountains in the Sierra Nevada Mountain Range (Fig. 3). This profile spans a wide range of topography, from grasslands on rolling hills,





to tall pine trees near Alta Sierra, see Supp. Fig. S5 for surface images along the profile. AMOG also
conducted a post-survey, downwind vertical atmospheric profile to 1800 m. Approximately 15 minutes of
data were collected in an open (200–300 m) field above Shirley Meadows (2258 m) that was fairly
exposed with only thin stands of pine trees on terrain falling steeply off to both sides.

The AMOG vertical ascent was collected before the AJAX profile to enable concurrent AMOG/AJAX
data collection for the Kern Fields. The AMOG ascent/descent was from 18:48 to 21:09 (20:08 UTZ at
crest), while AJAX flew a descent pattern from 20:58 to 21:04 UTC. The AMOG descent was shortened
to ~1000 m altitude (Glenville, CA) to allow AMOG to reach the Kern Fields nearly concurrent with
AJAX and GOSAT.

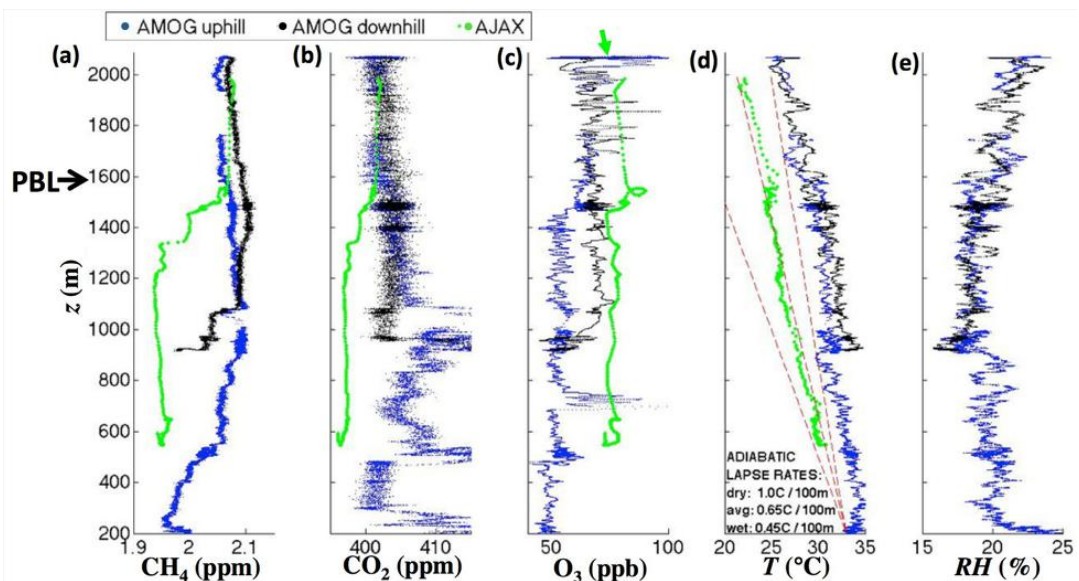


**Figure 4.** Surface altitude ($z$) profiles for west-east Delano-Alta Sierra transect (Fig. 3A, α–α') for
AMOG and AJAX (a) methane ($CH_4$), (b) carbon dioxide ($CO_2$), (c) ozone ($O_3$), (d) temperature ($T$), and
(e) relative humidity ($RH$). Also shown on (d) are the dry, average, and wet adiabatic lapse rates. Data key
on panel, planetary boundary layer (PBL), labeled. Green arrow shows extrapolation of AJAX trend to
Shirley Meadows altitude (2258 m).
Overlapping AMOG and AJAX profile data were collected between 500 and 2000 m. There was very
good agreement between the two platforms for $CO_2$ and $CH_4$ for altitudes between 1.55 and 2 km (Fig. 4a
and 4b). AMOG and AJAX $CH_4$ concentrations decreased notably from the well-mixed PBL to the near
surface layer, from ~2.07 ppm (500-750 m) to ~1.93 ppm (250-300 m). AJAX also showed a decrease in
CO2 from 403 ppm to below 400 ppm. The $CO_2$ decrease was consistent with a shift to agricultural air





where $CO_2$ vegetative uptake reduces $CO_2$ concentrations. The PBL grew from 600 to 900 m between
AMOG's ascent and descent and then to 1500 m by the time of AJAX's descent based on the $CH_4$, $CO_2$,
and $O_3$ data.

The PBL was identified at ~1580-1600 m based on both surface and airborne relative humidity (*RH*) and
temperature (*T*) vertical profiles. Diurnal heating is apparent between the two AMOG Surveyor *T* profiles,
but does not change the lapse rate. Because AJAX flies above the surface where AMOG collects data,
AJAX temperatures are lower. In the lower atmosphere, the lapse rate was 6.9°C km$^{-1}$ for AJAX between
500-900 m, while the AMOG lapse rate from 200-900 m was a similar 5.6°C km$^{-1}$. Between 950 and the
top of the PBL, AMOG lapse rates were much shallower, 2.5 °C km$^{-1}$, with a jump in temperature at 900
m. Above the PBL, the AMOG lapse rate was 3.5°C km$^{-1}$, close to the wet adiabatic lapse rate (Fig. 4d).

Above the PBL, $O_3$ concentrations between AMOG and AJAX were ~20 ppb different although the
AMOG and AJAX profile slope ($dO_3/dz$) were the same. If the trend in AJAX $O_3(z)$ from 1600 to 1850 m
is extended to $z = 2258$ m (Fig. 3C, green arrow), there is agreement with AMOG Shirley Meadows (open
field) $O_3$ concentrations. This similar slope but different absolute value could indicate $O_3$ loss as it
diffused down through the pine canopy to the surface (and AMOG). Tall pine trees dominate above
~1700, except for Shirley Meadows where, as noted, there was agreement. This difference does not arise
from calibration differences; the AMOG Surveyor $O_3$ analyzer was cross calibrated with the AJAX
calibration source. For $900 < z < 1400$ m, AJAX - AMOG agreement was better for the descent, which
was closer in time to AJAX than the ascent. This shift likely was associated with formation of the daytime
PBL.

AJAX observed elevated $O_3$ that was well mixed down to 500 m, while earlier AMOG showed well-
mixed $O_3$ down to only 1100 m. There also was a small (~10 ppb) $O_3$ enhancement at the top of the PBL
in both the airborne and surface profiles. The highest $O_3$ concentrations were observed by AMOG in
Shirley Meadows, where visibility was low due to smoke aerosols from the Rough Fire (NASA, 2015).
Air above the PBL was more humid than elsewhere in the profile, except for the lowest 50 m above the
valley floor, which was enriched in $CH_4$, $CO_2$, and *RH*, possibly from nocturnal accumulation and
agriculture including irrigation *RH* inputs. There were thin, atmospheric layers that suggest remnant
structures from the prior day. For example at ~550 m the air changed character, with a jump in $CO_2$ by
~10 ppm, and of $O_3$ by ~ 10 ppb, and a decrease in the $CH_4$ altitude gradient ($dCH_4/dz$).





Air was more polluted at greater altitude above the PBL in the upwind (Delano – Alta Sierra) profile for
$O_3$ for both platforms with air 10-20 ppb greater than in the PBL. Additionally, AJAX $CH_4$ and $CO_2$ were
significantly higher above the PBL. The AMOG $CH_4$ and $CO_2$ data are less clear, presumably because
AMOG data were prior to the disappearance of the nocturnal, stably stratified PBL. This was consistent
with visual observations of haze by AMOG from Shirley Meadows as well as by the AJAX pilot. Also,
air above the PBL was more humid.

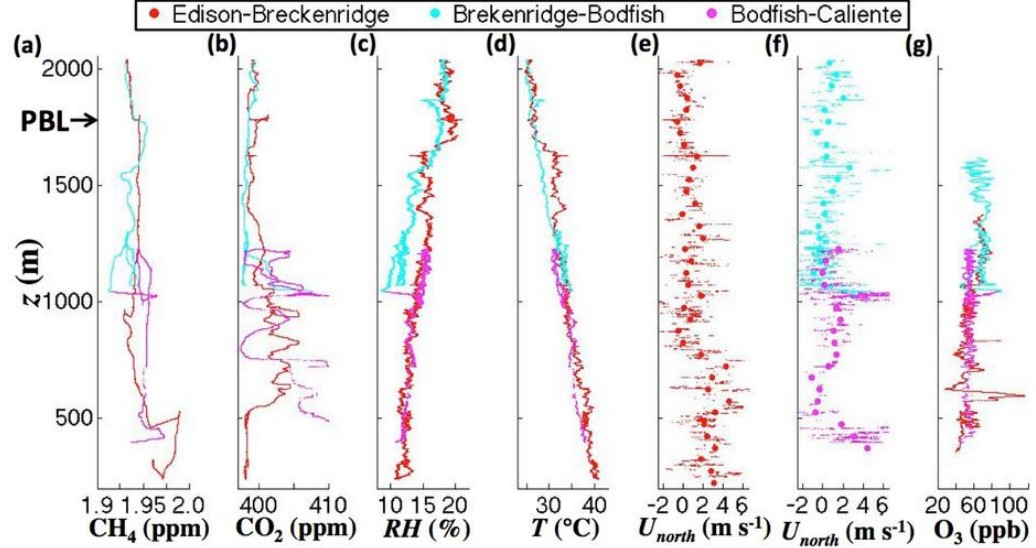


**Figure 5.** Surface altitude ($z$) profiles for Edison-Breckenridge ascent (red) and descent (blue) to Bodfish
and then Caliente profile (magenta) (Fig. 3b) for AMOG Surveyor (a) methane ($CH_4$), (b) carbon dioxide
($CO_2$), (c) relative humidity ($RH$), (d) temperature ($T$), north wind ($U_{north}$), for (e) ascent and (f) descent,
and (g) ozone ($O_3$). Planetary Boundary Layer (PBL) labeled.
A downwind ascent profile in the SJV was collected from Edison, CA to the high flanks of Breckenridge
Mountain, followed by a descent behind the Breckenridge Mountain to Caliente, CA through the tiny
town of Bodfish (Fig. 3b). This descent was separated from the SJV by a ridge and includes dryer, clean
air is that is representative of air from around Lake Isabella, a fairly isolated mountain valley. The
downwind profile was collected quasi-Lagrangian in that the time separating the two profiles (about four
hours) is comparable to the transport time (75 km at 4 m s$^{-1}$, implies 5 hours for transport). Thus, the
downwind profile was for close to the same air. Over these hours, there was some additional PBL
development, ~100 m to ~1675 m, with highly uniform $CH_4$ between 1000 m and the top of the PBL (**Fig.**
**5a**). Thus, the PBL remained fairly stable over the course of the study. Air in both the upper PBL and



above was cleaner with lower humidity and $CH_4$ concentrations. Unfortunately, the $O_3$ analyzer
overheated during the ascent and resumed collecting data on the descent at ~1500 m.

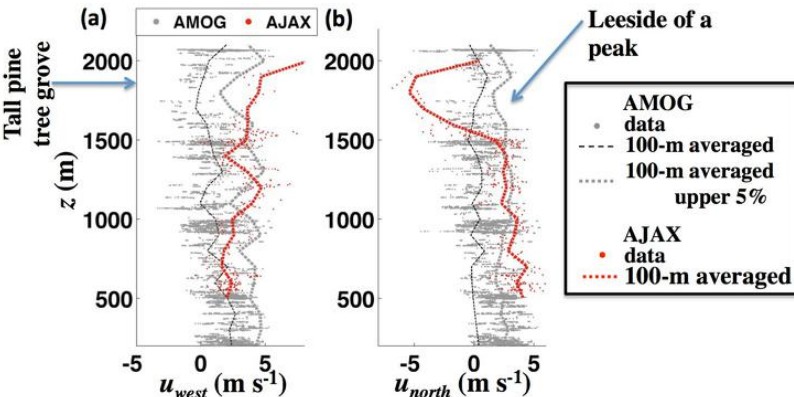


**Figure 6.** Altitude ($z$) profiles for (a) west (upslope) and (b) north (cross slope) wind components from
AMOG and AJAX for overlapping altitudes of the Delano–Alta Sierra transit (Fig. 3, α–α'), 100-m
altitude rolling-averaged data for AJAX, AMOG, and AMOG upper 5% of winds. Data key on figure.
Direct comparison between AMOG and AJAX winds is inappropriate because AMOG winds are affected
strongly by obstacles including hills, trees, and buildings. However, in many instances, terrain is open, or
gently rolling hills, and there tend to be regions of stronger winds that we propose are representative of
free atmosphere winds. AMOG data were altitude binned and the strongest winds in each bin were
compared with AJAX (Fig. 6). Agreement is generally good (within 15-20%) between the upper 5% of
AMOG cross-slope (west) winds in each altitude-averaged band (Fig. 6a). For the upslope wind (north)
agreement is better (within 5-10%) for a larger range of altitudes (Fig. 6b). This allows fusions of the
upper 5% of AMOG winds with AJAX winds.
**3.2. Kern Fields and Bakersfield Greenhouse Gas Emissions**
**3.2.1 Methane**
On 19 Aug. 2015, winds over the Kern Fields were prevailing (northwesterly) and fairly strong (~3 m s$^{-1}$)
on the ground and somewhat stronger aloft (Fig. 7). As a result, surface topography like the Kern River
Bluffs imposed only small wind modification at the surface and at altitude. Southeast of Bakersfield,
winds veered to westerly's towards passes in the Sierra Nevada Mountains that connect to the Mojave
Desert. The downwind survey included two plume transits on agricultural roads with negligible to no
traffic. These transits clearly showed the plume's eastward drift, passing to the north of the small town of
Arvin, CA.





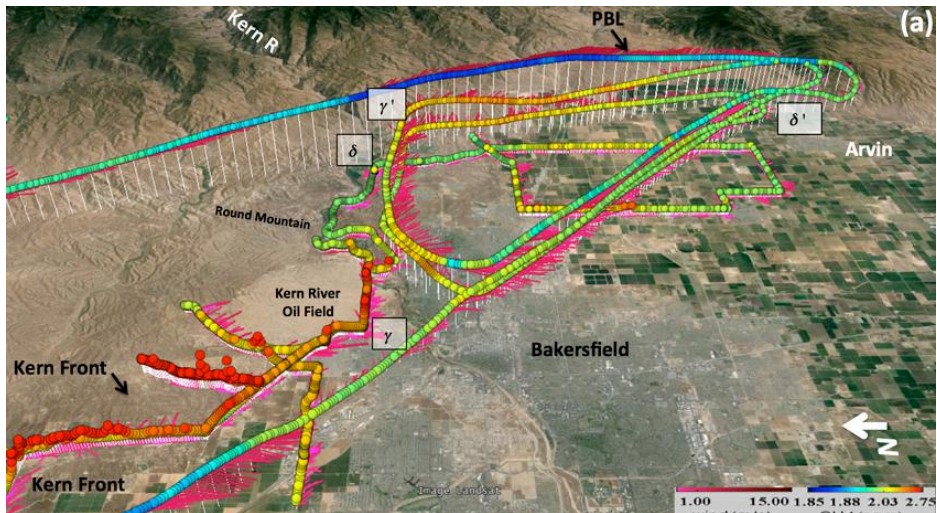


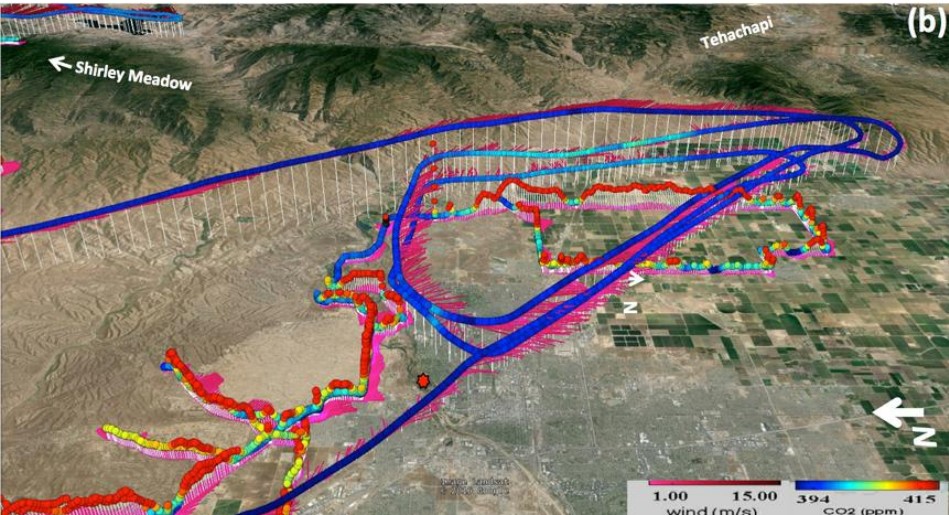


**Figure 7.** Combined AJAX and AMOG winds and *in situ* (a) methane (CH$_4$) and (b) carbon dioxide
(CO$_2$) for the Kern Fields on 19 Aug. 2015 for prevailing wind conditions. Greek letters identify two
downwind curtains. Red star on (b) locates origin for transect γ–γ'. Data keys on figure.



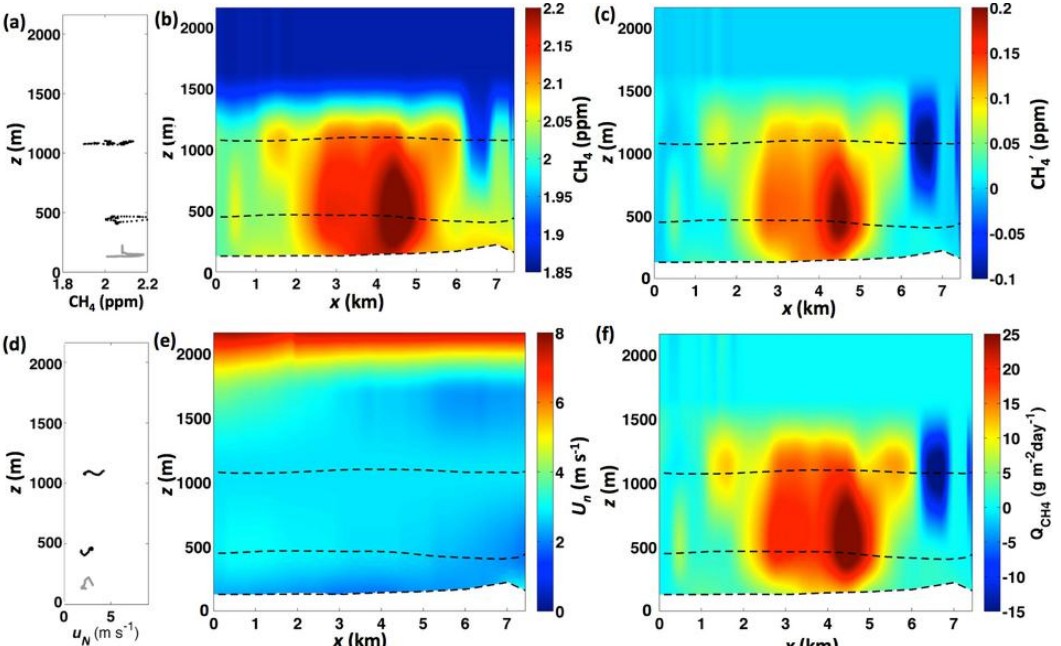


**Figure 8.** (a) Methane ($CH_4$) altitude ($z$) profiles for 19 Aug. 2015 for AJAX (black) and AMOG (gray)
data. (b) Interpolated, fused AJAX and AMOG $CH_4$ data, with respect to lateral east distance ($x$) relative
to 119.0023°W, 35.3842°N for data plane γ−γ' (Fig. 7). Dashed lines show data locations. (c) $CH_4$
anomaly ($CH_4'$) relative to the background data plane (Supp. Fig. S6A). (d) Vertical normal wind profile
($U_n$) from AJAX (black) and AMOG (gray) data during ascent/descent, (e) interpolated, fused $U_n$, and (f)
$CH_4$ flux ($Q_{CH4}$) for the Kern Fields. Data key on panels.
The background $CH_4$ plane $C_B(x,z)$ was extracted from the $CH_4$ data outside the plume − $C_{BL}(z)$ and
$C_{BR}(z)$, see Eqn. (1) − immediately downwind of the Kern Fields (transect **γ−γ'**). $C_B$ showed a slight
increase towards the east of ~20 ppb (Supp. Fig. S6a). Emissions from the Kern Fields' were dominated
by a large, focused $CH_4$ plume (or group of plumes) in the core of a much broader, dispersed, and poorly
defined plume. This structure is evident in both surface AMOG data and in the lowest AJAX altitude for
plane γ−γ' with both showing the strongest peak at $x = 4.5$ km (Fig. 8b, dashed lines). Within the plume,
concentrations are elevated at altitude relative to the surface, indicating buoyant rise. The upper AJAX
flight line was several hundred meters below the top of the PBL (at ~1580 m, Fig. 4) and constrains the
main plume, which was centered in the PBL. Concentrations above the PBL were determined from AJAX
descent and ascent data (Fig. 4), which agreed with AMOG observations above the PBL. These
observations show that the plume was not well mixed across the PBL. Two other small plumes were
observed at $x \sim 1.7$ and 5.7 km that were not mirrored in surface data and were centered at the top of the



PBL, indicating strong buoyant rise within the PBL. The upper altitude clean air intrusion at $x$~6.5 km lies
downwind of Round Mountain Canyon to the east of the Kern River oil field (Fig. 8b, Fig. 7a for
location), but did not penetrate down to 500 m. The normal wind ($U_n$) was fairly uniform across the data
plane, including downwind of the canyon (Fig. 8e). Thus, the $CH_4$ flux ($Q_{CH4}(x, z)$ shows similar spatial
patterns to $CH_4$'($x$, $z$). Total estimated emissions ($E$) were 32±16 Gg yr$^{-1}$.

For comparison, a recent bottom-up estimate of $CH_4$ emissions based on production data for the Kern
Fields estimated 10-40 Gg $CH_4$ yr$^{-1}$ (68% Confidence Level), by combining oil and gas production data
with US-EPA emissions factors for associated wells (Jeong et al., 2014). Other $CH_4$ sources are unlikely
to confuse this interpretation as petroleum system emissions are ~20 times larger than estimated nearby
livestock and landfill $CH_4$ emissions of ~2.3 and 1.4 Gg yr$^{-1}$, respectively (Calgem, 2014).
**3.2.2. Carbon Dioxide**
Background $CO_2$ for data curtain γ–γ' (Supp. Fig. S6b) was highly uniform. Given the strong crosswinds
and care taken to avoid trailing other vehicles on the low-trafficked China Loop Road, these data passed
quality review–$CO_2$ exhaust contamination manifests as a dramatic increase in the standard deviation as
AMOG intersects a turbulent vehicle exhaust plume. There was a shallow $CO_2$ layer constrained to the
lower 100 to 200 m with ~10 ppm enhancement (Fig. 9a), also observed in the $CO_2$ vertical profile (Fig.
4b), a layer that was characterized by elevated relative humidity. Further evidence that these broad spatial
$CO_2$ emissions are real is from the spatial similarity to $CO_2$ enhancements in the lowest AJAX flight data
(Fig. 9c). For example the surface $CO_2$ plume was strongest at $x$~4.5 km in AMOG and AJAX data. The
broad spatial extent of these emissions, similar to the broad $CH_4$ emissions suggests a relationship to
field-scale (engineering or geological) processes. Overall $CO_2$ emissions were 2.4±1.2 Tg yr$^{-1}$.

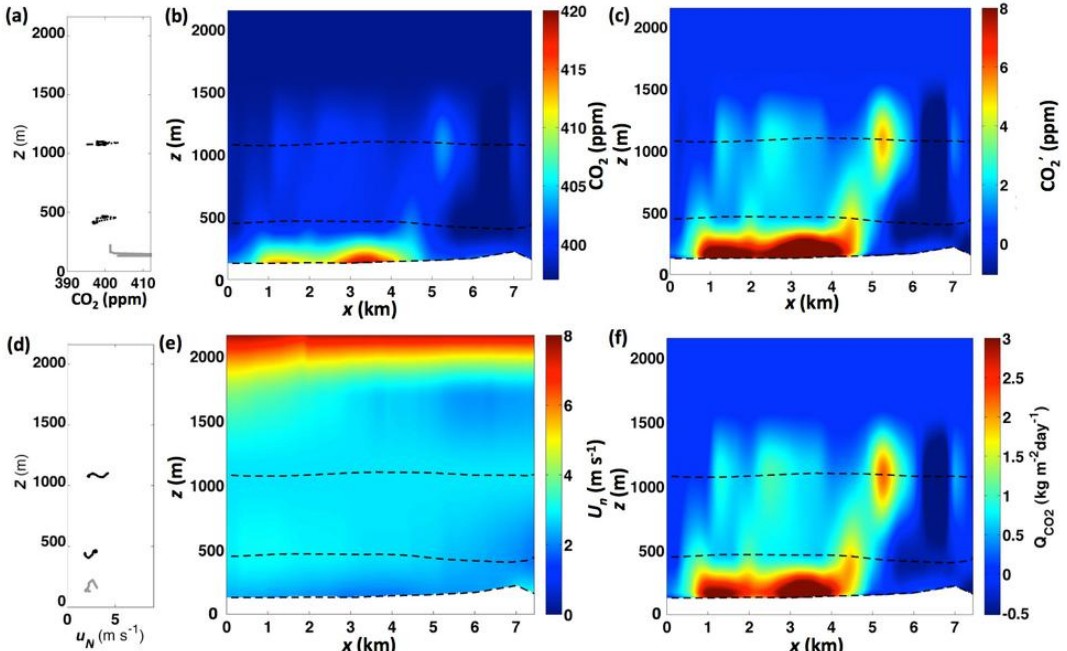

**Figure 9.** (a) Vertical carbon dioxide ($CO_2$) altitude ($z$) profile data for 19 Aug. 2015 for AJAX (black) and AMOG (gray) data. (b) Interpolated, fused AJAX and AMOG $CO_2$ data curtain with respect to lateral east distance, $x$, relative to 119.0023°W, 35.3842°N for curtain γ−γ' (Fig. 7b). Dashed lines show data locations. (c) $CO_2$ anomaly ($CO_2$'). (e) Vertical normal wind profile ($U_n$). (e) Interpolated, fused $U_n$, and (f) $CO_2$ flux ($Q_{CO2}$) for the Kern River and Kern Front oil fields for 19 Aug. 2015. Data key on panels.

There was a strong $CO_2$ anomaly in a focused plume at $x = 5$ km and $z = 1$ km. This plume likely relates to the two cogeneration power plants located in the Kern River oil field. Further support for this interpretation is its co-location with a similarly focused $CH_4$ plume at the same location. This power plant-related feature is a persistent feature that has been observed in other surveys (Leifer – unpublished data). The upper clean air intrusion in the $CH_4$ data curtain also is apparent in the $CO_2$ data (Fig. 9b), in front of Round Mountain Canyon (Fig. 7).

Based on a reservoir $CO_2$:$CH_4$ gas ratio of 92.2%:1.7% (Lillis et al., 2008) and 32 Gg yr$^{-1}$ $CH_4$ emissions, the Kern Fields' $CO_2$ emissions were predicted to be 1.8 Tg yr$^{-1}$, which is fairly consistent with the directly derived emissions of 2.4 Tg yr$_{-1}$. Both these values are somewhat lower than the inventory for the cogeneration plants in Kern River oil field - 3.1 Tg yr$^{-1}$ (CARB, 2016). The disagreement with inventory likely arises from intermittent activity, which was observed during the GOSAT-COMEX campaign.



## 4. Discussion

### 4.1. Experimental design and real-time visualization

Ideally, GCE airborne and surface data are collected first upwind and then downwind. However, AJAX
airborne data are not collected Lagrangian as would be necessary for slower, less maneuverable airborne
platform thanks to its extreme speed and maneuverability. This allows collection of near snapshot (~30
minutes) data. Slower, AMOG surface data were collected quasi-Lagrangian, reducing the likelihood of
confounding interference in the study area from non-FFI SJV inputs due to wind shifts after the pre-
survey (for non-nominal winds the collection is aborted). Given the AJAX-AMOG speed difference,
concurrent surface and airborne data could not be collected both upwind and downwind, and thus,
concurrency was prioritized for downwind. For flight efficiency and to provide downwind concurrency
with AMOG, AJAX flew a triangle that allowed AJAX to complete transects at three altitudes in close to
AMOG's upwind-downwind survey time.
After the Kern Fields survey, AJAX returned to base, while AMOG collected additional surface data,
exploring the fate of emissions from the Kern Fields. The word, "exploring" is significant, as real-time
visualization of winds, $CH_4$, and $O_3$ guided the downwind surveying. Data were collected to test the
hypothesis that there was a relationship between wind strength and the specific outflow path from the SJV
to Mojave Desert - specifically, that more northerly passes, which require greater wind veering from
prevailing are preferred at lower winds speeds. The AMOG survey first confirmed that outflow was not
up the Kern River Valley, and then collected a downwind vertical profile into the Sierra Nevada
Mountains to search for outflow through a pass near Breckenridge Mountain. After confirming its
absence, AMOG then investigated in the Tehachapi Pass, where the outflow was identified. Thus, on 19
Aug. 2015, when winds were strong, the outflow was by the most direct pathway - the Tehachapi Pass.

### 4.2. Experimental design and uncertainty reduction

The experimental design reduced uncertainty by better characterizing the PBL through surface and
airborne data fusion so that a well-mixed PBL is not required. Airborne data characterizes $CH_4$ and winds
in the PBL and above, while surface data characterizes the atmosphere below where airplanes are
permitted to fly due to airspace restrictions, e.g., cities, approach pathways, military airspace, and/or
safety. The *in situ* analyzers record concentration and winds with very high accuracy; however, only at a
single location and time. Thus, *in situ* uncertainty arises mostly from inadequate characterization of
temporal variability and spatial heterogeneity.






Aerial survey altitudes were designed to span from near the top of the PBL to as low as permissible and
an intermediate level (0.5, 1, 1.2 km). Thus, surface data added information on the lowest third of the
1.58-km thick PBL. This lower portion of the PBL is more important on days when the PBL is shallower.

For a well-mixed PBL, surface - airborne data fusion does not reduce uncertainty; however, these
observations showed that the well-mixed PBL assumption often may be poor (even 10-20 km downwind).
One solution is to collect data even further downwind, where the PBL is more well-mixed (White et al.,
1976); however, secondary (potentially uncharacterized) sources downwind of the study area and upwind
of the downwind data plane add confounding anomalies. Also, wind flow complexity can lead to transport
orthogonal to the overall downwind direction, leading to flux leakage out of the plume. The likelihood of
plume loss increases over greater distances. And finally, as the PBL evolves with time, it imposes an
evolving structure on the wind and concentration vertical profiles, which also challenge the well-mixed
PBL assumption – particularly if transport to the downwind plane requires hours.

Uncertainty also arises from wind and emission variability over the survey time period. The best strategy
is to minimize study time; however, there is a necessary tradeoff between spatial resolution and study
time. AJAX collects data quickly, allowing survey completion within far less than typical atmospheric
change timescales. Similarly, the surface survey route was designed to minimize collection time,
primarily on rural/agricultural roads carefully selected to avoid traffic congestion and traffic lights. The
surface survey requires ~90 minutes to complete and is conducted quasi-Lagrangian.

GCE treats uncertainty explicitly, allowing improvements in the data collection strategy to reduce
uncertainty. For example, the east-west downwind transect was lengthened from earlier data collects to
characterize background concentrations better. GCE also does not require an *a priori* emission
distribution and thus incorporates explicitly emissions from super-emitters, normal emitters, and
distributed sources, improving robustness of the findings. In contrast, inversion models require a
reasonable spatial *a priori* emission distribution and the ability to model transport across the study
domain. However, complex wind flows from fine-scale topographic structures, as observed for the Kern
Fields, challenge transport modeling.
**4.3. Profile intercomparison**
Above the PBL, there was excellent agreement between surface and airborne concentration profile data,
while concentration profiles within the PBL show significant differences between the two profiles, likely



related to air mass shifts and diurnal heating during the time between the profiles (Fig. 4). Winds above
the PBL were in poor agreement, with the north component in the opposite direction (Fig. 6). Underlying
this discrepancy was a mountain peak, which clearly caused large-scale alterations in the wind flow field.

Within the PBL, agreement between unfiltered surface AMOG winds and AJAX winds was poor,
unsurprising because surface winds are strongly affected by obstacles. However, by filtering AMOG
winds (collected 3-m above the surface) for the strongest 5%, agreement was within 15-20% for the
along-slope – i.e., north – winds, and better for upslope winds (west). Specific exceptions were when
AMOG was in a dense grove of pines, and when AJAX flew behind into the lee of a mountain peak.
Surface winds are modulated by a wide range of surface factors including trees, steep hills and hillocks,
blocking by a steep slope, rolling hills, and structures (Supp. Fig S5). However, a combination of gusts
(among thin wooded terrain on steep slopes) and the limited spatial extent of most obstacles underlies the
agreement between the filtered AMOG and AJAX wind profiles. Agreement is better for the upper
portions of the PBL (within 10-20%) where Sierra Nevada Mountain slopes are steeper. In contrast, the
slope lower in the PBL is gentle, and surface boundary layer effects are more pronounced, biasing wind
speeds slower.

The wind orientation to the slope affects the comparison because topography imposes wind field structure
at large and small scales. Where winds advect air upslope, transport incorporates a non-negligible vertical
component that is missed by the 2D sonic anemometer used in the study reported here. The current
AMOG configuration measures 3D winds, as does AJAX.

Some of the discrepancy between AMOG and AJAX wind profiles could have arisen from temporal
changes between the two profiles; however, this is unlikely for two reasons. First, the top of the PBL was
identified four times over the course of the study and remained stable within 100 m across the domain.
And second, surface wind observations remained relatively constant after the mid-morning shift to
daytime conditions (breakup of nocturnal stratification). However, the poor agreement between AJAX
and AMOG vertical concentration profiles within the PBL suggests significant air mass shifts –
highlighting the need for better concurrence.
**4.4. GHG FFI emissions**
Emissions for the Kern Fields were estimated at $32\pm16$ Gg $CH_4$ yr$^{-1}$ with $CH_4$ emissions ~20% above
EPA inventories, and $2.4\pm1.2$ Tg $CO_2$ yr$^{-1}$. The broad $CO_2$ plume suggests emissions from the geologic
reservoir – likely along the same pathways associated with $CH_4$ leakage – in addition to the focused





emissions from the co-generation power plants. On China Loop Road (where the $CO_2$ surface plume was
transected), strong crosswinds and light traffic would have prevented significant vehicular $CO_2$
contamination.

For comparison, a recent bottom-up estimate of $CH_4$ emissions from the Kern Fields estimated 25±15 Gg
$CH_4$ yr$^{-1}$ by combining oil and gas production data with emissions factors for associated wells used by
US-EPA (Jeong et al., 2014). Thus, 19 Aug. 2015 $CH_4$ emissions were a third above inventories. A
number of factors likely play a role including the age of the Kern River oil field (over a century),
production factors (steam injection), shallowness of the reservoir (<300 m), location in a tectonically
active area, which creates alternate migration pathways from the reservoir (Leifer et al., 2013), and the
recent expansion of the number of wells in the Kern Front oil field (from GoogleEarth timeline imagery).
Many of these factors are common to other production fields in California, the US, and globally.

These results agree with a recent metastudy of field studies of FFI production emissions, which showed
significant underestimation in the EPA budget (Brandt et al., 2014; Miller et al., 2013). Given the
importance or dominance of FFI emissions in anthropogenic greenhouse gas budgets, an increase of 25-
50% of the FFI contribution requires either reduction in another budget category, and/or an increase in the
loss rate. However, a recent husbandry emissions study also suggested significant underestimation
(Gentner et al., 2014). Thus, the present study supports the hypothesis that $CH_4$ loss rates are
underestimated. For example a recent study identified a new loss mechanism in near-surface soils
(Fernandez-Cortes et al., 2015). In any case, this study highlights the need for improved measurement
tools to reduce the significant uncertainty in the $CH_4$ budget and also satellite measurement validation,
particularly for complex terrain and in the source's near field. Mountainous terrain affects about half the
earth's population and half the earth's surface (Meyers and Steenburgh, 2013).
**5. Conclusion**
This study showed how to combine airborne and surface *in situ* data to improve emissions derivation, and
demonstrated the novel use of topography to characterize vertical atmospheric structure with a surface
mobile platform. Given that mountains cover a significant fraction of the earth's land surface, and that
airplane logistics often are beyond the available resources for many researchers, there are many
opportunities to apply these techniques globally. Data showed the PBL was not well-mixed, even 10-20
km downwind, highlighting the importance of the direct flux quantification experimental design.



**Table of Nomenclature**

| | Units | Description |
|---|---|---|
| | Units | Description |
| AJAX | (-) | Alpha Jet Atmospheric eXperiment |
| AMOG | (-) | AutoMObile trace Gas |
| Bbl | (-) | Barrel (of oil) 1 bbl = 6.38 m$^3$ |
| COMEX | (-) | CO2 and MEthane eXperiment |
| EOR | (-) | Enhanced oil recovery (techniques) |
| EPA | (-) | Environmental Protection Agency |
| GCE | (-) | GOSAT COMEX Experiment |
| GHG | (-) | Greenhouse Gases |
| GOSAT | (-) | Greenhouse gases Observing SATellite |
| GHG | (-) | Greenhouse gas |
| PBL | (-) | Planetary Boundary Layer |
| SJV | (-) | San Joaquin Valley |
| Tg | | Terragram ($10^{12}$ g) |
| UTZ | (-) | Universal time |
| $C'(x,z)$ | (ppm) | concentration anomaly (above $C_B$) |
| $C(x,z)$ | (ppm) | concentration |
| $C_B(x,z)$ | (ppm) | background concentration – outside plume |
| $C_{BL}(z)$ | (ppm) | background concentration profile – left side of profile |
| $C_{BR}(z)$ | (ppm) | background concentration profile – right side of profile |
| $E$ | (mol s$^{-1}$) | Emission source strength |
| $N'$ | (mol cm$^{-3}$) | molar mass anomaly |
| $Q(x,z)$ | (mol m$^{-2}$ s$^{-1}$) | Flux through the data plane |
| $R^2$ | (-) | Correlation coefficient |
| $RH$ | (%) | Relative humidity |
| $T$ | (°C) | Temperature |
| $U_n(x,z)$ | (m s$^{-1}$) | Winds normal to the data plane, a function of (x, z) |
| $U_{north}$ | (m s$^{-1}$) | North wind component |
| $U_{west}$ | (m s$^{-1}$) | West wind component |
| $x$ | (m) | lateral distance – approximately cross-wind |
| $x_L$ | (m) | left half of the transect ($x<x_{max}/2$) |
| $x_{max}$ | (m) | length of a transect |
| $x_R$ | (m) | right half of the transect ($x>x_{max}/2$) |





| 600 | *y* | (m) | lateral distance – approximately co-wind |
| 601 | *z* | (m) | altitude |
| 602 | $\Phi_L(C)$ | (-) | concentration probability distribution for left side of transect |
| 603 | $\Phi_R(C)$ | (-) | concentration probability distribution for right side of transect |
| 604 | $\Phi_P(C)$ | (-) | concentration probability distribution for the plume |
| 605 | $\Phi_B(C)$ | (-) | concentration probability distribution for the background |
| 606 | α, α' | (-) | designation for Delano – Alta Sierra surface transect |
| 607 | ε, ε' | (-) | designation for Edison– Breckenridge Mtn. surface transect |
| 608 | τ, τ' | (-) | designation for Breckenridge – Caliente surface transect |
| 609 | β, β', $β_1$' | (-) | designation for Wasco – Granite surface transect |
| 610 | γ, γ' | (-) | designation for Oildale – Oil City surface and airborne transects |
| 611 | δ, δ' | (-) | designation for Ming Park – Arvin surface and airborne transects |


**Data Availability.** Data will be provided as per the data policy.

**Author Contribution.** I. Leifer prepared the manuscript with input from all co-authors. C. Melton
prepared figures and conducted data analysis. M. Fischer helped prepare the emissions budgets. J. Frash
helped with AMOG data collection. L. Iraci, J. Marrero, J-M. Ryoo, T. Tanaka, and E. Yates are part of
the AJAX team and worked to collect and analyze AJAX data.
There are no competing interests

**Acknowledgements:** We thank the NASA Earth Science Division, Research and Analysis Program, grant
NNX13AM21G. MLF was supported by a grant from the California Energy Commission's Natural
Gas Research Program to the Lawrence Berkeley National Laboratory under contract DE-AC02-
36605CH11231. AJAX data were collected under the AJAX project, which acknowledges the partnership
of H211, LLC and support from the Ames Research Center Director's funds.

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
