# Peer review of "Atmospheric Characterization through Fused Mobile Airborne"

_Atmospheric Measurement Techniques, 2017_

## Referee Comment (RC1) · Anonymous Referee #1 · 26 Jun 2017

Comments on Leifer et al: Improved atmospheric characterization...

General comments: This is an interesting and innovative paper, decribing how both mobile surface measurement across strong local topography and parallel aircraft measurements have been used to derive methane emission fluxes. This is an important problem, and has wide applications. By using aircraft and vehicle to validate each other, both methodologies are improved. In the US, instrumented aircraft are available. In many less wealthy nations that is not the case and measurement will have to depend on vehicles supplemented by light drones. The topographic problem is also

widely applicable, in the many locations where the boundary layer is poorly mixed. Thus although aspects of the experiment can be criticised and the introductory section is very inadequate, the paper is an interesting and useful contribution, that should be published after revision.

Specific details: The Introductory section is incoherent, and depends in part on outdated papers. It needs to be revised substantially.

Line 35 should probably cite up to date NOAA results, perhaps from Dlugokencky's papers. For lifetime (also line 43-44), be careful not to mix up the various definitions of lifetime and again best to use a recent Dlugokencky paper or use IPCC. Line 38 – Nisbet et al 2015 is an error. No such paper. It's 2014. Maybe better cite the major Nisbet et al 2016 Glob. Biogeo. Cycles paper, and also Schwietzke et al. (in Nature recently) on fossil fuel emission. Either here or in L50 cite Saunois, M. et al (2016) The global methane budget 2000–2012, Earth System Science Data, 8, 697-751, doi:10.5194/essd-8-697-2016. Line 43 Claim of a 40% drop in lifetime - There is a major discussion on whether or not OH is changing and maybe that's far beyond the scope of this paper. Also may be confusing lifetime definitions (perturbation/replacement) in general comment. Line 49. See for example Rigby, M. and 18 others (2017) Role of atmospheric oxidation in recent methane growth. Proc. Natl. Acad. Sci. USA. 114, 5373-5377. Line 50. Maybe cite Saunois, M. et al (2016) again. Line 51 on – several more recent papers to cite on US fossil fuel emissions. Bruhwiler et al (2017) in JGR very effectively questioned Turner et al. , so I think this could be rewritten as the Turner et al conclusions should be discounted. Line 58-59 – The inventory discussion could be picked up around line 55 and should lead into Kirschke et al, and Saunois et al (2 papers, 2016, 2017) and maybe mention Nisbet and Weiss's earlier top down/bottom up comment in Science, 2010. Line 62 – many more recent papers than White 1976! This paragraph L61-70 needs updating. Also it could mention some of the European work – for example in several papers by Bergamaschi et al. Line 92-3 could be written by a PR firm selling dodgy goods. Most people live on the plains, not in the high Himalayas. Line 120 – reference for air flow to 1km only? Figure 1a looks like a photo of some poster on the floor...terrible, key is unreadable. Why the angled projection??? Line 168 'stranded CH4 clouds' – we see these too, but don't call them stranded. They are not washed up on the beach (strand) – they are making their own independent progress as yet unmixed. They are important as they come from single event sources, which can be keys to the emissions from a gasfield – eg water processing. Line 196 - precision and accuracy please, calibration etc etc. (though I accept the point in L252) Lines 228-247 calculation and also 248-253. Maybe cite some of the Aliso canyon work here? L271 – pre/post-surveys. Good. When was this? How comparable in meteorology?

L300 ± 25 This section is very interesting. I'm surprised the PBL is located to a 20m precision in the complex topography. Seems very precise. L312 – tall pine trees? How tall - Not being Californian, are these 5m (tall in N.East Norway hills) or 150m redwoods? L342 – really a town called Bodfish? are they inland codfish? or an obscure mermaid-body genus? Is this paper really about Bodfish bubbleology? The mind boddles. L365 – there is a lot of agriculture in the area – cows, wet fields, drains. This needs to be discussed. Are there any large cattle feed lots in and around Bakersfield? Landfills? Fig 8 – really interesting figure could have a bit more discusssion. L396 – how rapid is the rise? Any guesses? L402 – nice to get some d13C(CH4) isotopes to pin down the methane. See comment above about cattle and landfills in the area. Also Bakersfield is a big city – sewage. L435 – intermittent activity – what the paper calls 'stranded plumes' – any idea how common this is? L440,442 – collected Lagrangian – English language disjunction. ?in Lagrangian mode?? L441 – poor English again – a platform? Or platforms? L471 – faux precision. 1.6km L473-480 – good. L503 – mountain peak – hitherto hidden in the discussion, now pops out. I hope you told the pilot...could be mentioned earlier as a problem. L532 – maybe cite some of the NOAA work in other gasfields – intermittent sources. L533 – CO2 – maybe, or just a lot of trucks and all sorts of other sources? L540 and 551 – would be nice to see some isotopes on this. L556 – repeat comment – no, we do NOT all live on the tops of Everest

and Aconcagua. We live in the swamps of Calcutta and Shanghai. Schiphol runway, the center of Holland, is 11 ft under the ocean. Agreed, this comment is very useful for Tehran and La Paz (both with major local gasfields) but it's irritating to hammer it twice.

Overall – interesting paper. Accept with revision.

―――――――――――――――

---

## Short Comment (SC1) · 26 Jun 2017

There really is a town called Bodfish in the southern Sierra Nevada Mtns named after the Gold miner (George Bodfish)

tall pine trees are 50 m. See photo in supplemental Fig. S5.

We accept and will address the other comments.

---

## Referee Comment (RC2) · Anonymous Referee #2 · 6 Jul 2017

"Improved Atmospheric Characterization through Fused Mobile Airborne & Surface In Situ Surveys: Methane Emissions Quantification from a Producing Oil Field" Leifer et al.

Reviewer Comments:

This manuscript describes a campaign conducted using a surface mobile platform along with an airborne platform, to estimate CH4 and CO2 emissions from California's Kern Oil Field using one day of measurements. In principle, this is a valid attempt

to use two different platforms and merge the data sets to create a better picture of emissions, which would be worthwhile and could help improve current methodologies. But in practice, in my opinion, this manuscript does not accomplish that. For a submission to AMT, there is very little precision in the method description or the description of the measurements themselves. There is no quantification of how well the concentration measurements from the two platforms compare, nor is there any description of the interpolation method for the sparse measurements in space and time. The description of the flux calculation is confusing and the description of the uncertainty analysis is too short, especially given the focus of this journal on measurement techniques and methods. In the conclusion several statements are made that are, in my opinion, overly broad and have not been shown in the work, including the interpretation of the final emissions estimate in relation to a bottom-up inventory estimate and in relation to global trends. In addition, it is not clear to me even that this method works in such complex terrain and conditions (the authors do spend a large portion of the paper describing the terrain, which they use to their advantage) which the authors do note. It is certainly not shown that this method might be better than any other method in terms of accuracy - this may not be required for publication, but it is claimed by the authors. The manuscript would be improved by focusing on the methodology itself and justifying the various methods used to perform the estimate, and remove the focus on the estimated emissions result and the global methane budget.

Specific comments:

L 54-55 "most" should be "largest" L55 Does the EPA inventory discuss the global budget? What about agriculture? (The latest Global Methane Budget Saunois et al may be a better citation here, and they place agriculture and waste to be significantly larger contributor than fossil fuel, so this statement does not seem to be supported by the literature).

L56-60 - Turner and Bruhwiler disagree as to whether a methane trend is detectable over the US; the Bruhwiler paper specifically refutes the results in Turner. So the

phrasing here is not really correct, but could be phrased to simply emphasize that there is diagreement in the literature over whether a methane emissions trend in the US exists. More recent literature on the global methane is also available suggesting that the OH sink is the cause of the global increase (as mentioned in lines 41-46). References to a more recent Turner et al and Rigby et al, both in PNAS, 2017 should be made if this is to be discussed. [However, as noted above, perhaps devoting a large portion of the paper to the global methane budget is not really in the scope of this work].

L62 - Peischl, White and Karion all use esssentially the same method (mass balance) - not sure what is meant by "direct assessment". Perhaps because of the aircraft-based winds used in the first two while Karion relied on model or ground observations?

L69 - This discussion should include a citation to Smith et al, 2015, ES&T where the ethane/methane ratio was not assumed a priori but determined from the airborne data, but still used to apportion emissions in the Barnett.

Fig 1: a) Panel in figure is not clear - much too small to read. (b) North should be indicated.

L163-165 repetitive? awkward.

L167 CGE should be GCE?

Stranded plumes: This should be more clearly described as plumes that are coming into the domain from upwind or outside the domain. At least this seems like what is being described here, that there is a criterion of a "clean" or at least relatively uniform upwind condition.

L170: What is the specific criterion for "too light or variable" on the wind speeds? In my opinion, this is very subjective in the description, including "flush nocturnal accumulations before the overpass" - so this is a restriction on wind speed in terms of transit time?

L179: These studies or most of them used compressed gases as standards either on-board or prior and after the campaigns - calibration is still required with CEAS systems.

L193 At what height is the air sample drawn relative to the roof and the anemometer?

L184-205 Have the environmental variables been compared with local weather stations or other sensors for validation (i.e. of wind measurements, which as is noted in the text, can be difficult because of the need to account for vehicular motion)?

I was looking for this reference, which is cited for the instrumentation: "Leifer, I., Melton, C., Manish, G., and Leen, B.: Mobile monitoring of methane leakage, Gases and Instrumentation, July/August 2014, 20-24, 2014.", not clear what journal or no way to find this?

Information should be given in a similar fashion as they are in 2.3 for AJAX on calibration methods. (see later comments on the supplement).

When merging the two data sets for a single analysis it becomes even more important to show that measurements of methane are on the same scale relative to the same standards, etc. and have been intercompared. The vertical profile indicates that they compare "well" at high altitudes, but no quantity is given.

L230-231 $U\_n$ and $U\_N$ are both representing perpendicular winds.

Section 2.4: This is not clear enough. There should be an equation here for Q as a function of U and C - initially it is reported that Q is simply the product but later that C is converted to a mass (density?) to derive an emission rate. What are the units of Q (flux of what? grams or moles?). If the emission rate units are in moles, then this is not required (as in Cambaliza et al). The derivation of the background is also not very clear here - why it must be split into a right and left half? Might be more clear to describe x as the coordinate from the beginning of a transect to the end, and $x_{max}/2$ is the midpoint for each transect upwind? Is there no x-dependence of CBL and CBR? (L235 indicates they are only functions of z?). An example would be nice here.

– Just looking now at the list of definitions (thank you, this clarifies things!), and it becomes clear that when the authors refer to concentration they are actually referring to a mole fraction, i.e. micromoles of CH4 per mole of dry air (this should be defined), or ppm. Concentration is usually (if molar concentration) in units of moles per meter cubed (in SI units - the authors use mol per cm cubed), which could make it a "molar mass anomaly" for the authors (N'). These should be re-defined correctly in the future draft for section 2.4 - call C a mole fraction and N a concentration.

This section is unclear with equation (1) not clear to me why the integral of a gaussian distribution would be zero. Not clear how C' is related to Phi_P. A reader has to work way too hard to make sense of this method. Also, from Figure S6, it seems that Phi is a distribution for each transect, but in the equation C is a function of z. How is the vertical interpolation done, and how is C' defined?

Equation 2 does make sense, although x1 and x2 should be defined.

Figure S5: What is this figure telling us? What are the colors in the tracks (yellow/green?). Could an elevation map suffice here?

Figure S6: Is there a transect upwind at 2200 m (as in (c)?), but there is no dashed line in (a) corresponding to this one? The data should be shown as well as the interpolated curtain, to show if there is spatial structure in x and z of this background that is being smoothed?

Supplement S5: L168 should read section 2.4 in text. L167. not clear. So the peak of the distribution (is this the mode?, i.e. the value most commonly seen in this upwind transect?), is used as the value for the entire transect, and then the background was interpolated vertically - how? Is x in Fig 6 going from west to east?

L253, I agree, but wouldn't call this "appropriateness" - more specific. Maybe appropriateness of the measurements for the assumptions made for the analysis? What assumptions are being made that need to be satisfied? I would call it representativity

or just say that spatial and temporal variability are the dominant sources of uncertainty.

L276: to 1800 masl (from what base elevation?)

L277: at 2258 is this part of the profile? Isn't this higher then?

Fig 4 (indicate masl rather than just meters for clarity)

L314, this is nice to note, but should also be included in the supplementary section on measurements, as well as to what precision they agreed (within X ppb agreement on average, or something quantitative).

L325, don't remnant structures from the prior day make the mass balance or emissions estimate not correct, according to earlier text about flushing out prior days' emissions? (from reading on, we see this is the "upwind" profile, so this should be mentioned here or somewhere nearby).

L329 (alpha - alpha' should be used here for clarity for the reader).

Fig 4: For the upwind profile, alpha-alpha', the CH4 is lower in the PBL than above. However, in Fig S6, the upwind "curtain" or plane, is showing higher CH4 at lower altitudes. How are these two figures consistent?

In Figure S6, which of the transects are AJAX, and which AMOG?

Fig 5 why only is the north wind shown? These are very low wind speeds indeed, esp. for doing an emissions estimate.

L345 yes at 4 m/s - is that the wind speed? It's not shown. Was that the wind speed for 5 hours?

L347 how is growing from 100 to 1675m a stable PBL? Also, is stable referring to the atmospheric stability class (i.e turbulence) or the fact that the PBL depth is not changing much in time?

L368 Westerlies?

Figure 8: what is the time difference between when these transects were measured, as well as when the transects for the background (showin in Figure S6) sampled? Was the bacgkround plane subtracted point by point, i.e. in x, z space so that a higher background was subtracted on the east side, (L386)? Still don't understand where the distributions Phi come in to this picture.

Fig. 8 how was the vertical interpolation done, and the extrapolation above the highest flight transect at ~1100 masl? It seems like a different method was used for $U_n$ and CH4, noting where they drop off in the vertical. Figure 4 indicates that AMOG was driving the surface transect much earlier than the AJAX transects (or perhaps I misinterpreted this), so how can we combine them when we know the PBL is growing?

L400: Extrapolating these emissions to an annual average is a stretch and not at all defensible. This is one of the reasons that recent similar studies that are performed over a short time frame report their emission rates in moles per second or kg per hour or such. The section on the uncertainty estimate is short and not thorough - the distributions that go into the Monte Carlo would need to be explained better.

L431: Could you look at a slope of the CH4 to CO2 tracer plot in the plume to show this consistency with the reported ratio?

L496 I would say that these complexities also challenge this method because of the variability that you are not measuring - and the model you are using assuming some constancy in wind.

Overall, this method does not fully account or try to discount the possibility of unsteadiness in the winds between upwind and downwind transects that could lead to accumulation of emissions during slower wind speed periods (night time but also could just be earlier in the day). Perhaps this is dealt with in the uncertainty calculation but that is not clear in the text as written here.

L499-501, Please indicate some quantification of the differences here. This is a methods paper - how well did the concentrations (mole fractions) agree (above the PBL), in ppb? Were any calibration tanks measured on both systems?

L502-505, yes it is true that we could not expect the winds to agree, but what does this indicate for the interpolation of wind in the vertical from the different platforms? Is that variability captured at all?

L523+ What about plumes of CH4 that are following these complex winds and topography? the simple interpolation and treatment of the surface data is troublesome under these conditions.

A mass balance equation is a conservation of mass and the equations (although not written out here) assume some steady uniform wind condition. This is clearly violated here. Perhaps the uncertainty calculation deals with this problem but it is not clear.

L541: Are these factors not accounted for in the inventory? What about temporal variability? Also, what about the uncertainties on both numbers, assuming they are 1-sigma (which should also be noted incidentaly)? Seems to me the emissions estimate actually overlaps with the inventory quite well given the uncertainties that are reported.

L552. In my opinion, this should be toned down - this one measurement supports the conclusion is that the global loss rate of CH4 to OH (or soils) is underestimated? What percentage of the global methane budget does 25 or 32 Gg/year actually represent?

Conclusion

L559. This statement implies that the uncertainty has been reduced from other methods, which is not the case, and has not been shown.

L562 But this method relies on the aircraft measurements as well as the surface, so could not be applied in the absence of those resources!

L564 - The flux quantification is "direct", meaning measured winds and concentrations were used, but that is the flux through a point in space and time $(Q(x,z))$ - the rest is

a simple model: you must integrate that flux based on an interpolation (in space and time), and must subtract a background that has its own model and interpolation, and the attribute that flux to a surface emission which requires some Eulerian conditions - steady flow through a control volume. All are a "model" - just a simple one.

However this point can be made differently - that one should measure before adding one more assumption to the model, which is that of a vertically well-mixed plume. Other studies have moved away from this assumption of vertical well-mixedness as well: Cambaliza et al., Heimburger et al. (Elementa 2017), Lavoie et al. (ES&T 2015 and 2017), Conley et al (both 2016 as well as 2017: http://www.atmos-meas-tech-discuss.net/amt-2017-55/), and numerous others, especially when sampling in the near field. I agree that this is a valid point to make using these observations.

Supplement:

L26 cfm should be given in metric

What are some estimated uncertainties on the FGGA CH4 measurements based on the calibration standard - how often is it sampled, si there noise/drift, etc? A sentence or two on this is warranted beyond just the statement that a calibration was performed. Was there a water correction, or were the dry values reported by the FGGA used?

The additional accuracy of the 450C sentence should go where it is first discussed, before the sentence about the FGGA. Earlier it says it achieves 1ppb accuracy, but now it says that it can achieve 50ppt if calibrated with hourly zero gas mesaurements - which number applies here? Where do the authors get the accuracies reported for the other analyzers (ozone, etc)? Manufacturer? If the main paper is not about these auxiliary gases, this information should not really be mentioned and could be removed. Interestingly, no accuracy or uncertainty is reported for CO2 or CH4, the main gases of interest in this work (for the AMOG measurements).

S2.2: Is there a reference for the MMS wind system? There is no information given

here, and this is a key measurement for flux studies. Uncertainty on winds should be reported for both platforms.

---

## Author Response (AR1)

Comments on Leifer et al: Improved atmospheric characterization:

General comments: This is an interesting and innovative paper, describing how both mobile surface measurement across strong local topography and parallel aircraft measurements have been used to derive methane emission fluxes. This is an important problem, and has wide applications. By using aircraft and vehicle to validate each other, both methodologies are improved. In the US, instrumented aircraft are available. In many less wealthy nations that is not the case and measurement will have to depend on vehicles supplemented by light drones. The topographic problem is also widely applicable, in the many locations where the boundary layer is poorly mixed. Thus although aspects of the experiment can be criticized and the introductory section is very inadequate, the paper is an interesting and useful contribution, that should be published after revision.

Specific details: The Introductory section is incoherent, and depends in part on outdated papers. It needs to be revised substantially.

Line 35 should probably cite up to date NOAA results, perhaps from Dlugokencky's papers. For lifetime (also line 43-44), be careful not to mix up the various definitions of lifetime and again best to use a recent Dlugokencky paper or use IPCC.

*Yes, Ed's work now cited.*

Line 38 – Nisbet et al 2015 is an error. No such paper. It's 2014. Maybe better cite the major Nisbet et al 2016 Glob. Biogeo. Cycles paper, and also Schwietzke et al. (in Nature recently) on fossil fuel emission. Either here or in L50 cite Saunois, M. et al (2016) The global methane budget 2000–2012, Earth System Science Data, 8, 697-751, doi:10.5194/essd-8-697-2016.

*We have cited Saunois et al 2016 and Nisbet et al 2016. There was an error in the endnote database for Nisbet 2014 that has been corrected.*

Line 43 Claim of a 40% drop in lifetime - There is a major discussion on whether or not OH is changing and maybe that's far beyond the scope of this paper. Also may be confusing lifetime definitions (perturbation/replacement) in general comment.

*We agree that lifetime, although worthy of mentioning in general, is well beyond the scope of the paper and thus the lifetime aspects are now greatly reduced.*

Line 49. See for example Rigby, M. and 18 others (2017) Role of atmospheric oxidation in recent methane growth. Proc. Natl. Acad. Sci. USA. 114, 5373-5377.

*Now cited in the OH loss paragraph with a summary sentence.*

Line 50. Maybe cite Saunois, M. et al (2016) again.

*Done*

Line 51 on – several more recent papers to cite on US fossil fuel emissions. Bruhwiler et al (2017) in JGR

very effectively questioned Turner et al. , so I think this could be rewritten as the Turner et al conclusions should be discounted.
Deleted.

Line 58-59 – The inventory discussion could be picked up around line 55 and should lead into Kirschke et al, and Saunois et al (2 papers, 2016, 2017) and maybe mention Nisbet and Weiss's earlier top down/bottom up comment in Science, 2010.

Line 62 – many more recent papers than White 1976!
True, but they really do not make this point as specifically or as well. Also, much of the science of my father's generation seems to be in the process of disappearing down the memory well – and so I would like to keep White highlighted. That said, the works of Peischl et al. (2015; 2016) is added. I note that both Peischl et al., (2015;2016), also cite White et al. (1976).

This paragraph L61-70 needs updating. Also it could mention some of the European work – for example in several papers by Bergamaschi et al.

Have added Saunois et al. (2017), who is working these days with Bergamaschi on inversion modeling.

Line 92-3 could be written by a PR firm selling dodgy goods. Most people live on the plains, not in the high Himalayas.

Point taken. Deleted. Although most people lived on the plains, wherever feasible, the wealthy have always lived on more defensible hills above the plains where the poor lived (and were flooded, marauded, etc.

Line 120 – reference for air flow to 1km only?
Zhong et al., (2004) again. Added.

Figure 1a looks like a photo of some poster on the floor: : :terrible, key is unreadable. Why the angled projection???
Redone

Line 168 'stranded CH4 clouds' – we see these too, but don't call them stranded. They are not washed up on the beach (strand) – they are making their own independent progress as yet unmixed. They are important as they come from single event sources, which can be keys to the emissions from a gas field – e.g., water processing.

Agreed, they are not stranded on a beach. Rephrased as "upwind $CH_4$ plumes"

Line 196 - precision and accuracy please, calibration etc etc. (though I accept the point in L252)
Added.

Lines 228-247 calculation and also 248-253. Maybe cite some of the Aliso canyon work here?
Done. Thanks!

L271 – pre/post-surveys. Good. When was this? How comparable in meteorology?
Added. "Primary changes were development of near surface winds, and a slight increase in the PBL"

L300-25 This section is very interesting. I'm surprised the PBL is located to a 20-m precision in the complex topography. Seems very precise.

Its sharp – and something that we often see in California. The Marine layer PBL is often visualized by clouds and is equally sharp (almost every day on my morning commute), sometimes to a few meters!

L312 – tall pine trees? How tall - Not being Californian, are these 5m (tall in N.East Norway hills) or 150m redwoods?

They are tall in a California sense. Added that they are 30m+

L342 – really a town called Bodfish? are they inland codfish? or an obscure mermaid-body genus? Is this paper really about Bodfish bubbleology? The mind boddles.

It was a gold rush town named after a gentleman last name bodfish. The history of California is weird!

L365 – there is a lot of agriculture in the area – cows, wet fields, drains. This needs to be discussed. Are there any large cattle feed lots in and around Bakersfield? Landfills?

THe one nearby dairy in the upwind direction is now pointed out on fig. 7A, and discussed. Potential plumes from the only nearby upwind dairy (Fig. 7a, white arrow) were directed by winds to pass to the west of the oil fields. The only Landfill is to the south (and not along the flow path. There are no wet fields in this part of the San Joaquin Valley – that is only in the Sacramento area. Water is a little too precious here, and it gets too hot!

Fig 8 – really interesting figure could have a bit more discussion.

L396 – how rapid is the rise? Any guesses?

I think the answer is pretty darn fast, methane has half the density of air, but mixing is not going to be similar to that for a 600K plume. In the Aliso Canyon leak (see https://www.youtube.com/watch?v=exfJ8VPQDTY) it looks like many meters per second. That said, it seems a bit overly speculative to put a number in print….

[Figure]

L402 – nice to get some d13C(CH4) isotopes to pin down the methane. See comment above about cattle and landfills in the area. Also Bakersfield is a big city – sewage.

Agreed, however, no isotope instruments were available.

L435 – intermittent activity – what the paper calls 'stranded plumes' – any idea how common this is?

For CO2, this would be more from cycling of the co-generation power plant being on and off, the language is now clarified to "the co-generation plant only being active some of the time, confirmed by data from the GOSAT-COMEX campaign."

L440,442 – collected Lagrangian – English language disjunction. ?in Lagrangian mode??

Changed to "in a Lagrangian sense"

L441 – poor English again –a platform? Or platforms?

Changed to "measurement platforms"

L471 – faux precision. 1.6km

Changed. Thank you, that is something I usually catch in reviews I do!

L473-480 – good.

L503 – mountain peak – hitherto hidden in the discussion, now pops out. I hope you told the pilot: : : could be mentioned earlier as a problem.
The pilot flies more or less where we ask, but of course they can change their route as they see fit. Also, they are military pilots flying a converted military jets, and tend to do things their way a bit, For example, in one of the flight segments (not here), the pilot realized he needed. Now discussed in section 2.1

L532 – maybe cite some of the NOAA work in other gasfields – intermittent sources.
Apologies, but the intermittency for CO2 was from cycling the power plan on and off. Actually, and this is the subject of another paper that we are writing, on the field level, CH4 emissions are not intermittent, but stochastic – law of large numbers of leaks. In that paper, though, we do show that at the well level, they are intermittent. The text has been adjusted slightly.

L533 – CO2 – maybe, or just a lot of trucks and all sorts of other sources?
Thanks, this needed extra clarification. Added "Additionally there are no upwind (non-oil field) roads, only the foothills of the Sierra Nevada Mountains."

L540 and 551 – would be nice to see some isotopes on this.
Agreed, however, no isotope instruments were available.

L556 – repeat comment – no, we do NOT all live on the tops of Everest and Aconcagua. We live in the swamps of Calcutta and Shanghai. Schiphol runway, the center of Holland, is 11 ft under the ocean. Agreed, this comment is very useful for Tehran and La Paz (both with major local gasfields) but it's irritating to hammer it twice.
Having lived a year in Den Haag, I understand your point. Deleted.

Overall – interesting paper. Accept with revision.

Improved Atmospheric Characterization through Fused Mobile Airborne & Surface In Situ Surveys: Methane Emissions Quantification from a Producing Oil Field" Leifer et al.

Reviewer #2 Comments:

This manuscript describes a campaign conducted using a surface mobile platform along with an airborne platform, to estimate CH4 and CO2 emissions from California's Kern Oil Field using one day of measurements. In principle, this is a valid attempt to use two different platforms and merge the data sets to create a better picture of emissions, which would be worthwhile and could help improve current methodologies. But in practice, in my opinion, this manuscript does not accomplish that. For a submission to AMT, there is very little precision in the method description or the description of the measurements themselves. There is no quantification of how well the concentration measurements from the two platforms compare, nor is there any description of the interpolation method for the sparse measurements in space and time.

The description of the flux calculation is confusing and the description of the uncertainty analysis is too short, especially given the focus of this journal on measurement techniques and methods.

In the conclusion several statements are made that are, in my opinion, overly broad and have not been shown in the work, including the interpretation of the final emissions estimate in relation to a bottom-up inventory estimate and in relation to global trends.

In addition, it is not clear to me even that this method works in such complex terrain and conditions (the authors do spend a large portion of the paper de- scribing the terrain, which they use to their advantage), which the authors do note. It is certainly not shown that this method might be better than any other method in terms of accuracy - this may not be required for publication, but it is claimed by the authors.

The manuscript would be improved by focusing on the methodology itself and justifying the various methods used to perform the estimate, and remove the focus on the estimated emissions result and the global methane budget.

• We agree with this assessment and plan to address flux and global implications in a future paper. Thus, we have deleted the discussion as recommended by the reviewer. We have clarified our argument that the approach is better and under what conditions, based on the philosophy that reducing extrapolation improves outcomes. We now devote the first paragraph of section 4.2 to this important issue. Additionally, we have re-arranged section 4.2 to remove some duplication. Also, we have toned down the title.

• WRT to the approach being better – our contention is simple – more information or more complete characterization of the PBL means less interpolation and extrapolation which cannot help but reduce uncertainty for any method. As to whether the improvement is significant or trivial, that of course depends on the situation. In any case, the text no longer makes the claim.

• With respect to leveraging terrain, I fully accept it would be better if I had an airplane and could fly and collect data. But most scientists on this planet very seldom have access to an airplane, particularly on a regular basis, and as a result, by leveraging terrain, scientists like myself, can collect useful vertical atmospheric profile data. In the San Joaquin Valley, agreement between the airplane data and car data was very good over a wide range of altitudes indicating that at a minimum, the method has significant promise. We have added a paragraph at the beginning of section 4.3 that now recommends further research and highlights that it may not be appropriate elsewhere.

• Based on the comments from the reviewer, we decided to add a paragraph to the methodology that explains why we are using an anomaly approach rather than an upwind/downwind mass balance approach. See the paragraph, which is presented below. Since this is key, a short summary now also appears in the study motivation section.

> • "Unfortunately, the upwind data showed a lateral gradient, which coupled with uncertainty in precisely where the downwind air originated (given the topography, which features a gentle incline towards the northeast, this gradient is unsurprising, in retrospect). Thus a very small shift in the winds between the upwind and downwind curtains results in a significant shift in $C_B$, with a very large effect on $Q$. As a result, the more traditional upwind/downwind mass balance approach was abandoned for an anomaly approach."

Specific comments:

L 54-55 "most" should be "largest" L55 Does the EPA inventory discuss the global budget? What about agriculture? (The latest Global Methane Budget Saunois et al may be a better citation here, and they place agriculture and waste to be significantly larger contributor than fossil fuel, so this statement does not seem to be supported by the literature).

• As this is background material, we have added to the current text the Saunois reference to highlight that there is uncertainty in the CH4 budget.

L56-60 - Turner and Bruhwiler disagree as to whether a methane trend is detectable over the US; the Bruhwiler paper specifically refutes the results in Turner. So the phrasing here is not really correct, but could be phrased to simply emphasize that there is disagreement in the literature over whether a methane emissions trend in the US exists. More recent literature on the global methane is also available suggesting that the OH sink is the cause of the global increase (as mentioned in lines 41-46). References to a more recent Turner et al and Rigby et al, both in PNAS, 2017 should be made if this is to be discussed. [However, as noted above, perhaps devoting a large portion of the paper to the global methane budget is not really in the scope of this work].

• We agree and delete the sentence as ancillary to the main focus of the study.

L62 - Peischl, White and Karion all use essentially the same method (mass balance) - not sure what is meant by "direct assessment". Perhaps because of the aircraft-based winds used in the first two while Karion relied on model or ground observations?

• Yes, the difference is whether transport (i.e., transport) are measured in the study domain, or need to be modeled correctly. The sentence has been re-written to clarify what is meant by direct assessment.

L69 - This discussion should include a citation to Smith et al, 2015, ES&T where the ethane/methane ratio was not assumed a priori but determined from the airborne data, but still used to apportion emissions in the Barnett.

• Agreed. We missed Smith et al 2015 due to the long time frame between writing the paper and submission – the intro material was written in 2015! We have also added a citation to the more recent Peischl et al., 2016, too. In the same vein, we have added a citation to a recent paper by Schwietzke et al. 2016 to bring the introduction more up to date.

Fig 1: a) Panel in figure is not clear - much too small to read. (b) North should be indicated.

• Replaced by a higher resolution figure. North is now indicated on panel b.

L163-165 repetitive? awkward.

• Thanks, rewritten less repetitive

L167 CGE should be GCE?

• Thanks, yes. Fixed.

Stranded plumes: This should be more clearly described as plumes that are coming into the domain from upwind or outside the domain. At least this seems like what is being described here, that there is a criterion of a "clean" or at least relatively uniform upwind condition.

• Stranded plumes are plumes that due to wind shifts are no longer contiguous to their source (at least in the transit type of data we collect). Stranded plumes have been observed frequently in the SJV by AMOG.

• That said, whether they are stranded or connected to their source, is not relevant to whether they can disrupt the experiment, so the term stranded is dropped.

L170: What is the specific criterion for "too light or variable" on the wind speeds? In my opinion, this is very subjective in the description, including "flush nocturnal accumulations before the overpass" - so this is a restriction on wind speed in terms of transit time?

• Added detailed criteria. Winds speeds typically less than ~2 m s$^{-1}$, and variability less than 30°. Flush the nocturnal (i.e., no $CH_4$ cloud at or nearby upwind of the site, which means that winds could not have been light as recently as several hours prior; however, winds are not measured several hours prior),

L179: These studies or most of them used compressed gases as standards either on- board or prior and after the campaigns - calibration is still required with CEAS systems.

• True, and we do calibrations daily. Text reflected to note that the calibration gas does not need to be onboard the platform – it can be at the lab / hotel/ base site, etc.

L193 At what height is the air sample drawn relative to the roof and the anemometer?

• Sample is drawn from between 2 m above the roof and 3 m above the roof, depending on speed. This has been clarified by rewriting the paragraph.

L184-205 Have the environmental variables been compared with local weather stations or other sensors for validation (i.e. of wind measurements, which as is noted in the text, can be difficult because of the need to account for vehicular motion)?

• Pressure: We have compared with the Bakersfield airport, and agreement is within our uncertainty on the altitude of the height of the sensor, and pressure changes over an hour (the airport reporting time).

• Winds are much more challenging they are always changing and always spatially varying. Our best efforts have been to compare short data sets collected at a range of speeds on an open road, early in the morning, to compare wind measurements for driving the road in both directions (with the wind at an angle). Worst performance is at around 45 mph for winds of less than 1 m/s where errors are on the order of 20% in speed and direction. At higher wind speeds and/or lower driving speeds, error decreased rapidly. The GPS correction to real speed error is much smaller as it corrects itself after a few readings, which we distribute across the data by spatial filtering that limits accelerations in the along travel direction to physical limits and to near zero in the direction transverse to the direction of travel. Discussion with Vaisala, indicate that there is no need to send their wind sensor in for annual calibration (barring it being hit by a large branch). We also optimized the wind sensor positioning to minimize uncertainty for winds within ~30° from the front of the vehicle. Since the relative wind always has a very strong along travel direction component, this criteria is almost always met for driving at all but the slowest speeds and/or the strongest cross winds. We have not spent effort at looking at accuracy for very high cross wind data (have measured to 17 m/s) because we do not analyze such collected data in our studies to date. Additionally, such strong winds in California tend to be very strongly modified by topography – making them particularly challenging to validate.

• This above discussion has been added to the supplementary material.

I was looking for this reference, which is cited for the instrumentation: "Leifer, I., Melton, C., Manish, G., and Leen, B.: Mobile monitoring of methane leakage, Gases and Instrumentation, July/August 2014, 20-24, 2014.", not clear what journal or no way to find this?

• This paper is in a trade journal (it was peer reviewed) and is attached.

Information should be given in a similar fashion as they are in 2.3 for AJAX on calibration methods. (see later comments on the supplement).

• Done. We have also added our linear cell pressure calibration to the supplemental material.

When merging the two data sets for a single analysis it becomes even more important to show that measurements of methane are on the same scale relative to the same standards, etc. and have been intercompared. The vertical profile indicates that they compare "well" at high altitudes, but no quantity is given.

L230-231 $U_n$ and $U_N$ are both representing perpendicular winds.

• Thanks. Typo corrected.

Section 2.4: This is not clear enough. There should be an equation here for Q as a function of U and C - initially it is reported that Q is simply the product but later that C is converted to a mass (density?) to derive an emission rate. What are the units of Q (flux of what? grams or moles?). If the emission rate units are in moles, then this is not required (as in Cambaliza et al).

• Equation added, units added, and it is noted that there is a conversion factor between ppm and moles.

The derivation of the background is also not very clear here - why it must be split into a right and left half? Might be more clear to describe x as the coordinate from the beginning of a transect to the end, and xmax/2 is the midpoint for each transect upwind? Is there no x-dependence of CBL and CBR? (L235 indicates they are only functions of z?). An example would be nice here.

• It is split into a right and a left half due to gradients across the field. Since CBL and CBR are average values, they do not have an x dependency

– Just looking now at the list of definitions (thank you, this clarifies things!), and it becomes clear that when the authors refer to concentration they are actually referring to a mole fraction, i.e. micromoles of CH4 per mole of dry air (this should be defined), or ppm. Concentration is usually (if molar concentration) in units of moles per meter cubed (in SI units - the authors use mol per cm cubed), which could make it a "molar mass anomaly" for the authors (N'). These should be re-defined correctly in the future draft for section 2.4 - call C a mole fraction and N a concentration.

• An equation has been added, and text now notes that $C$ is in ppm and that there is a conversion factor to moles per volume.

• When there is a gradient as there almost always is in nature, it is unclear as to which part of the background is transported to the measurement plane, introducing uncertainty. We address this by derive background from the downwind dataplane.

This section is unclear with equation (1) not clear to me why the integral of a Gaussian distribution would be zero. Not clear how C' is related to Phi_P. A reader has to work way too hard to make sense of this method. Also, from Figure S6, it seems that Phi is a distribution for each transect, but in the equation C is a function of z. How is the vertical interpolation done, and how is C' defined?

• Equations have been added and the section has been clarified. Phis is a distribution for each left or right transect at each altitude (hence the z dependency). Interpolation, prior to integration, is linear, now stated

Equation 2 does make sense, although x1 and x2 should be defined.

• Equation 2 was rewritten to not use x1 and x2

Figure S5: What is this figure telling us? What are the colors in the tracks (yellow/green?). Could an elevation map suffice here?

• Data key added. The purpose of this figure is to show the typical surface obstacles to surface winds along the profile, elevation is not particularly relevant.

Figure S6: Is there a transect upwind at 2200 m (as in (c)?), but there is no dashed line in (a) corresponding to this one? The data should be shown as well as the interpolated curtain, to show if there is spatial structure in x and z of this background that is being smoothed?

• Since 2200 m is background, there is no need to separate the background concentration from a plume and it is used without smoothing or analysis in the linear interpolation.

Supplement S5: L168 should read section 2.4 in text. L167. not clear. So the peak of the distribution (is this the mode?, i.e. the value most commonly seen in this upwind transect?), is used as the value for the entire transect, and then the background was interpolated vertically - how?

• There was a mistake in figure S6 caption – the probability distributions are all for CH4, half for the right side of the data field and half for the left side of the data field. This has been corrected. Additionally, the methodology of filling in the background data plane is now also described in the supplemental material.

Is x in Fig S6 going from west to east?

• The caption for Fig S6 has been improved to include the definition of $x$ and $z$, as in the main text.

L253, I agree, but wouldn't call this "appropriateness" - more specific. Maybe appropriateness of the measurements for the assumptions made for the analysis? What assumptions are being made that need to be satisfied? I would call it representativity or just say that spatial and temporal variability are the dominant sources of uncertainty. L276: to 1800 masl (from what base elevation?)

• The is a good suggestion. Changed to represntative and to spatial and temporal variability. Also changed to 1800 masl.

L277: at 2258 is this part of the profile? Isn't this higher then?

• It was a typo – corrected to 2058 m here and elsewhere. This was above the airplane profile. Data were collected in the open to compare with the direction and speed of winds near the top of the overlapping profiles where AMOG was surrounded by tall trees, and showed good agreement This clarification now is added.

Fig 4 (indicate masl rather than just meters for clarity)

• Done.

L314, this is nice to note, but should also be included in the supplementary section on measurements, as well as to what precision they agreed (within X ppb agreement on average, or something quantitative).

• Moved to section 2.2 where calibration also is mentioned for the GHG analyzer.

L325, don't remnant structures from the prior day make the mass balance or emissions estimate not correct, according to earlier text about flushing out prior days' emissions? (from reading on, we see this is the "upwind" profile, so this should be mentioned here or somewhere nearby).

• The upwind profile is to characterize the atmospheric structure, not to provide input to the mass balance, and air from the profile will pass to the east of the oil fields. That this is the upwind profile is now mentioned.

L329 (alpha - alpha' should be used here for clarity for the reader).

• This would require doing so for all greek letters in all captions or the entire text, which seems excessive. We will discuss with the copy editor.

Fig 4: For the upwind profile, alpha-alpha', the CH4 is lower in the PBL than above. However, in Fig S6, the upwind "curtain" or plane, is showing higher CH4 at lower altitudes. How are these two figures consistent?

• The upwind profile is at alpha-alpha', Supp. Fig. S6 shows the background curtain (not upwind) curtain, correctly labeled in the caption and section heading, and is at delta-delta' and is derived from are data outside the plume. The methodology is described in Section 2.4, now corrected in the supplemental to refer to the right section.

In Figure S6, which of the transects are AJAX, and which AMOG?

• AMOG is on the surface, now labeled.

Fig 5 why only is the north wind shown? These are very low wind speeds indeed, esp. for doing an emissions estimate.

• As noted above, the profile is not for directly estimating emissions, but to characterize the atmospheric profile structure, primarily the location of the PBL. The figure was very busy already, so only the north component (ascent and descent) are shown. The purpose of showing the winds is to note they are not (in this case, useful for identifying the PBL, and that they show a change between the earlier and later profiles. The east component conveys similar (lack of useful) information.

• Text now notes "Winds were not useful for deriving the location of the PBL."

L345 yes at 4 m/s - is that the wind speed? It's not shown. Was that the wind speed for 5 hours?

• Yes – text clarified.

L347 how is growing from 100 to 1675m a stable PBL? Also, is stable referring to the atmospheric stability class (i.e turbulence) or the fact that the PBL depth is not changing much in time?

• For clarity, text changed to "~100 m growth to ~1675 m,"

L368 Westerlies?

• Changed.

Figure 8: what is the time difference between when these transects were measured, as well as when the transects for the background (showin in Figure S6) sampled? Was the background plane subtracted point by point, i.e. in x, z space so that a higher background was subtracted on the east side, (L386)? Still don't understand where the distributions Phi come in to this picture.

Fig. 8 how was the vertical interpolation done, and the extrapolation above the highest flight transect at ~1100 masl? It seems like a different method was used for U_n and CH4, noting where they drop off in the vertical. Figure 4 indicates that AMOG was driving the surface transect much earlier than the AJAX transects (or perhaps I misinterpreted this), so how can we combine them when we know the PBL is growing?

L400: Extrapolating these emissions to an annual average is a stretch and not at all defensible. This is one of the reasons that recent similar studies that are performed over a short time frame report their emission rates in moles per second or kg per hour or such. The section on the uncertainty estimate is short and not thorough - the distributions that go into the Monte Carlo would need to be explained better.

• Agreed. Results are now reported in Mol s-1 (with the Gg yr-1 reported in parentheses for comparison to inventory). We have also expanded the Monte Carlo approach section

L431: Could you look at a slope of the CH4 to CO2 tracer plot in the plume to show this consistency with the reported ratio?

• Its really the anomaly, and we are comparing the emissions, which accounts for this in a way that a scatterplot would not.

L496 I would say that these complexities also challenge this method because of the variability that you are not measuring - and the model you are using assuming some constancy in wind.

Overall, this method does not fully account or try to discount the possibility of un- steadiness in the winds between upwind and downwind transects that could lead to accumulation of emissions during slower wind speed periods (night time but also could just be earlier in the day). Perhaps this is dealt with in the uncertainty calculation but that is not clear in the text as written here.

L499-501, Please indicate some quantification of the differences here. This is a methods paper - how well did the concentrations (mole fractions) agree (above the PBL), in ppb? Were any calibration tanks measured on both systems?

• Calibration tanks for the two platforms were different, but from the same vendor. Agreement was 99.7% for $CH_4$ and 99.9% for $CO_2$. Comparison of the median winds showed is 38% for Ueast is 27%. A figure of the altitude variations is shown as Supp. Fig. S7.

L502-505, yes it is true that we could not expect the winds to agree, but what does this indicate for the interpolation of wind in the vertical from the different platforms? Is that variability captured at all?

• As noted in the text, this disagreement above the PBL is due to AJAX data being influenced by being collected in the lee of a mountain peak. This paragraph is onluy for winds above the PBL, The next paragraph is for winds in the PBL, where there was good agreement.

L523+ What about plumes of CH4 that are following these complex winds and topography? the simple interpolation and treatment of the surface data is troublesome under these conditions.

A mass balance equation is a conservation of mass and the equations (although not written out here) assume some steady uniform wind condition. This is clearly violated here. Perhaps the uncertainty calculation deals with this problem but it is not clear.

• The upwind profile and similarly the downwind profile are not for the purpose of characterizing the upwind or background concentrations, but solely for the purpose of identifying the PBL. We do compare extensively a range of parameters between AJAX and AMOG to show that surface profiles using topography can be used to characterize atmospheric structure. This is a second very important purpose of the manuscript, because not everyone has an airplane or access to an airplane, particularly in the developing world. We fully agree it is nice to have an airplane and I wish I had one of my own. Additionally, the upwind and downwind profiles are "inconveniently" located with respect to the study area (which is approximately 50 km to the south of the upwind profile. In the case of the flux calculation, stranded layers are not evident in the airborne profile data. This could be because they are more likely to occur in the mountains used for the surface profile, which is speculative, and thus is not mentioned in the manuscript. Given the typical air flow patterns, such features will typically not return to the SJV center when formed on the east mountains, but would for formation on the west mountains.

L541: Are these factors not accounted for in the inventory? What about temporal variability? Also, what about the uncertainties on both numbers, assuming they are 1- sigma (which should also be noted incidentally)? Seems to me the emissions estimate actually overlaps with the inventory quite well given the uncertainties that are reported.

• Text adjusted to note that the derived flux lies within the inventory uncertainty, but is higher, and then continues to note this is consistent with the metastudy of Brandt et al., that inventories likely are too low.

L552. In my opinion, this should be toned down - this one measurement supports the conclusion is that the global loss rate of $CH_4$ to OH (or soils) is underestimated? What percentage of the global methane budget does 25 or 32 Gg/year actually represent?

• Agreed, rewritten – see response above.

Conclusion

L559. This statement implies that the uncertainty has been reduced from other methods, which is not the case, and has not been shown.

• Removed

L562 But this method relies on the aircraft measurements as well as the surface, so could not be applied in the absence of those resources!

• Rewritten.

L564 - The flux quantification is "direct", meaning measured winds and concentrations were used, but that is the flux through a point in space and time ($Q(x,z)$) - the rest is a simple model: you must integrate that flux based on an interpolation (in space and time), and must subtract a background that has its own model and interpolation, and the attribute that flux to a surface emission which requires some Eulerian conditions - steady flow through a control volume. All are a "model" - just a simple one.

However this point can be made differently - that one should measure before adding one more assumption to the model, which is that of a vertically well-mixed plume. Other studies have moved away from this assumption of vertical well-mixedness as well: Cambaliza et al., Heimburger et al. (Elementa 2017), Lavoie et al. (ES&T 2015 and 2017), Conley et al (both 2016 as well as 2017: http://www.atmos-meas-tech- discuss.net/amt-2017-55/), and numerous others, especially when sampling in the near field. I agree that this is a valid point to make using these observations.

• Good points, and an additional sentence has been added to highlight.

Supplement:

L26 cfm should be given in metric

• Yes, done.

What are some estimated uncertainties on the FGGA CH4 measurements based on the calibration standard - how often is it sampled, is there noise/drift, etc? A sentence or two on this is warranted beyond just the statement that a calibration was performed. Was there a water correction, or were the dry values reported by the FGGA used?

• Added

The additional accuracy of the 450C sentence should go where it is first discussed, before the sentence about the FGGA. Earlier it says it achieves 1ppb accuracy, but now it says that it can achieve 50ppt if calibrated with hourly zero gas measurements - which number applies here? Where do the authors get the accuracies reported for the other analyzers (ozone, etc)? Manufacturer?

• This has been clarified. The accuracies are from the manufacturer and include 24 hour drift.

If the main paper is not about these auxiliary gases, this information should not really be mentioned and could be removed.

• We prefer to describe the system completely, and to include (with better explanation) the improvement in accuracy of the 450i by hourly zero measurement, as this could be of interest to other researchers

Interestingly, no accuracy or uncertainty is reported for CO2 or CH4, the main gases of interest in this work (for the AMOG measurements).

• Added

S2.2: Is there a reference for the MMS wind system? There is no information given here, and this is a key measurement for flux studies. Uncertainty on winds should be reported for both platforms.

• The MMS is a NASA developed system that has not been published. We provide a link to the homepage, and report its accuracies. Additionally, information on AMOG winds is now included, and explained as it depends on velocity of AMOG and the velocity of the winds.

[revised manuscript text omitted]

and hydrogen sulfide ($H_2S$). For all CEAS analyzers, dry values are used. Also, three chemiluminescence trace gas analyzers measure nitric oxide (NO) and nitrogen oxides ($NO_X$) at

0.1 Hz at 25 ppt accuracy (42TL, ThermoFischer Scientific, Waltham, MA), and ozone ($O_3$) at

0.25 Hz at 1 ppb accuracy (42C, ThermoFischer Scientific, Waltham, MA), and sulfur dioxide ($SO_2$) at 0.1 Hz at 1 ppb accuracy (450C, ThermoFischer Scientific, Waltham, MA). This accuracy is from the manufacturer and is based on 24 hour drift. Better accuracy is achieved by hourly zero gas measurements using chemically sparged air (Type CI, Cameron Great Lakes,

OH), which in the laboratory improved accuracy to 50 ppt. Given that $SO_2$ and $H_2S$ atmospheric concentrations are typically less than 1 ppb in California, this was an important improvement.

The FGGA is calibrated with an air calibration standard for greenhouse gases ($CH_4$: 1.981 ppmv;

$CO_2$: 404 ppmv; balance ultrapure air) and are stable to 1 ppb for CH4 over 24 hours, and 0.12

ppm for CO2 over 24 hours. Accuracy is <0.03%. Calibrations are performed before and after each field collect. The 49i was cross-calibrated with the AJAX $O_3$ analyzer to 1 ppb, and during a repeat cross calibration several months later had maintained its calibration to between 1 and 2

ppb.

Meteorology: A sonic anemometer (VMT700, Vaisala, Finland) is mounted 1.4 m above the roof and measures two-dimensional winds. Estimated accuracy is approximately 10° and 0.3 m s$^{-1}$ for wind speeds above 1.5 m s$^{-1}$; however, accuracy improves with vehicle velocity and wind speed as vehicle flow stream line interferences are reduced. Accuracy was determined empirically by driving several kilometers back and forth on a rural road in an open area in the early morning and comparing measured winds in the two directions. Note, these accuracies are greater than the manufacturer maximum error. At lower wind speeds, accuracy appears to be closer to 0.2 m s$^{-1}$, and 15-20°; however, is extremely challenging to determine. Still, afiltering, nocturnal wind data generally agrees well (~10°) with expectations from topographic forcing at wind speeds of ~0.2 -

0.5 m s$^{-1}$ on large spatial scales (tens of kilometers) for highway speed (140 km hr$^{-1}$) data. In general, winds are more accurate than stated if the winds are from within 30° of forward direction, as stated if they are from the side, unless strong (>~4 m s$^{-1}$), in which case they are equally accurate and very poor if from within ~15° of the behind direction. As a result, tail winds are not evaluated.

[revised manuscript text omitted]

$CH_4$ right side probability distributions (Φ).

[Figure]

**Figure S7** – Comparison between AMOG **(a)** $u_{north}$ **(b)** and $u_{west}$.

**Mobile Monitoring of Methane Leakage**

BY IRA LEIFER, PH.D., CHRISTOPHER MELTON, MANISH GUPTA, PH.D. AND BRIAN LEEN, PH.D.

**Abstract**

It is critical to monitor methane leakage from oil exploration and distribution pipelines for safety, climate-related issues, and profitability. Conventional monitoring technology, which involves collecting discrete air samples, provides neither spatial nor temporal resolution. With the recent advent of portable, laser-based analyzers, it is now possible to make real-time, high-fidelity mobile measurements of greenhouse gases and pollutants. In this paper, we describe the development of an AutoMObile greenhouse Gas Survey Platform (AMOG Surveyor) that includes trace gas analyzers, a global positioning system, a sonic anemometer, and a real-time data visualization package. Sample deployment data are presented to illustrate the ability of the AMOG Surveyor in identifying and locating natural gas leaks, and rapidly assessing their severity.

**Introduction**

With the dramatic increase in worldwide natural gas production and consumption, there has been strong, renewed interest in monitoring methane leakage for safety and climate-related issues as well as product loss and profitability. From years 2002 to 2012[1], there were approximately 800 significant natural gas pipeline incidents in the United States, including more than 250 explosions that killed over 100 people, injured over 450 others, and caused more than $800M in damages. The number of incidents is expected to increase as the distribution infrastructure continues to age while being stressed further by expanding domestic production from unconventional gas sources. In addition to safety issues, methane is a potent greenhouse gas with a global warming potential that is 21 times larger than carbon dioxide on a century timescale, and 100 times that of carbon dioxide on a decadal time-scale. In 2011, 69 billion cubic feet of methane is estimated to have leaked to the atmosphere in the U.S. alone from the natural gas distribution process[2] at a cost of approximately $1B, with other estimates suggesting leaked amounts and costs that are 10 times larger. Moreover, a recent study[3] suggests that the EPA may be underestimating methane emissions during the gas drilling phase by factors of 100 to 1000. This represents a significant impact on the bottom line, and may suggest that natural gas generated power would cause a greater climate impact than even that generated from coal burning, absent aggressive application of technology to prevent such leakages.

Traditionally, researchers have studied methane from leakage and other sources by either discrete, atmospheric flask samples, which are analyzed in the laboratory, or at a fixed measurement station with concentrations determined by gas chromatography coupled with a flame ionization detector (GC-FID). By their nature, such measurements are very limited in both spatial and/or temporal resolution.

Recently, with the advent of laser-based gas analyzers, continuous and rapid atmospheric methane measurements became possible, allowing researchers to measure temporal trends in methane concentrations. However, these analyzers are typically deployed in laboratories or environmental sheds, and, thus, still provide little to no spatial resolution. By correlating the methane readings with wind speed and direction (e.g., the eddy flux technique), a stationary analyzer can sometimes provide spatial resolution over a few square kilometers; however, more extensive spatial mapping is required to identify and characterize methane leakage, and fairly strict locale, emission, and wind conditions must be met. Thus, for all of these measurements, deconvolving the effects of temporal and spatial variability in source strength from temporal and spatial variability in transport is very challenging. Similarly, temporal variability, like shifting winds, pulsed emissions, or multiple shifting emission sites, invalidates underlying assumptions of eddy flux measurements.

**Development of a Mobile Monitor**

To get at the underlying emissions and sources, separating the temporal and spatial variability from transport is critical, and addressed by rapid mobile measurements – installing a methane analyzer in a mobile platform (e.g., car, boat, aircraft, etc…). As noted above, GC-FID is poorly suited for this application due to its slow analysis speed (e.g., minutes), need for consumable gases, vibration sensitivity, and maintenance requirements. The former is especially limiting, since, in a vehicle travelling 60± miles/hour, GC-FID provides spatial resolution of about a mile, useless for methane leak localization and mitigation, although useful for large-scale applications, like satellite validation[4]. Fast mobile spectral measurements, can provide the snapshot "image" of plumes of methane or other gases needed to detangle transport processes from the underlying leakage emissions. The mobile monitor described below combines an Off-Axis Integrated Cavity Output Spectroscopy (ICOS)[5], laser-based Fast Greenhouse Gas Analyzer (FGGA) with a global positioning system (GPS), sonic anemometer (to measure wind direction and speed), system diagnostic monitoring, and custom realtime data visualization software installed in a small commuter vehicle for real-time, spatially-resolved methane measurements. Secondary trace gases are measured by additional ICOS instruments, such as ammonia and $NO_2$, to discriminate between methane sources, such as dairies and combustion i.e., diesel trucks and other heavy machinery; key given that road measurements have a bias towards combustion emissions. The system is termed an AutoMObile greenhouse Gas Survey or AMOG Surveyor.

**Off-Axis ICOS Fast Greenhouse Gas Analyzer**

The Fast Greenhouse Gas Analyzer (FGGA) manufactured by Los Gatos Research is suited for the necessary rapid detection at high accuracy required by mobile platforms. The analyzer, shown in Figure 1, uses Off-Axis Integrated Cavity Output Spectroscopy (Off-Axis ICOS) to measure methane ($CH_4$), carbon dioxide ($CO_2$), and water vapor ($H_2O$) at 5 to 10 Hz. Such high speed is critical to obtain sufficient spatial resolution while moving at typical highway velocities in order to collect repeat data on quicker than typical atmospheric change times (hour-scale). The technique has been described in detail previously[6], and only a brief overview will be provided below.

[Figure]

*Figure 1. LGR Fast Greenhouse Gas Analyzer that utilizes Off-Axis ICOS to accurately quantify methane, carbon dioxide, and water vapor aboard a mobile platform at rates up to 10 Hz*

Two diode lasers operating near 1600 nm and 1650 nm for $CO_2$ and $CH_4$/$H_2O$ detection respectively are coupled into a high-finesse optical cavity consisting of two highly-reflective mirrors (R ≈ 99.99%). Light transmitting through the cavity is focused onto an amplified detector. A data control, acquisition, and analysis system tunes the lasers over a small spectral range (1 – 3 cm[-1]) at 300 Hz, digitizes the detector signal, averages 30–300 spectra (for 1–10 Hz response), and analyzes the cavity-

[Figure]

[Figure]

*Figure 2. AMOG Surveyor images at the California Polytechnic State University Dairy waste pool showing roof and trunk packages*

enhanced absorption spectra to determine the trace gas concentrations—in this case, $CH_4$, $CO_2$, and $H_2O$. A pump pulls sample through the cavity, while the cell pressure is actively regulated to attain high precision. By varying the pump flow, the analyzer can provide effective data rates ranging from 10–0.1 Hz, sufficient to meet both mobile and stationary (low power) monitoring applications.

In Off-Axis ICOS, the laser trajectory into the cavity is not critical and the resulting analyzer is not affected by small changes in optical alignment due to vibration, shock, and thermal stresses. The mirror coatings are made from metal oxides and do not degrade with time or chemical contact. Finally, although the gas sample is filtered, the mirrors can be removed, cleaned, and replaced in the field by a minimally trained operator if necessary.

**AMOG Surveyor**

The AMOG Surveyor is a Nissan Versa commuter car that has been modified for scientific trace gas surveys (see Figure 2). The Surveyor, which was developed based on tens of thousands of kilometers of data collection experience, has been designed to facilitate effective adaptive surveys for real-time, trace gas plume characterization. The Surveyor includes subsystems for power management, sample gas handling, gas analysis, thermal management, ancillary measurements, data communication, and real-time software.

**Power Management**

In order to obtain high spatial resolution, real-time data at high travel speed (e.g., 60 mph), the AMOG Surveyor utilizes a high-flow scroll pump (8 CFM or 30 CFM) to pull sample through the analyzers. This pump requires 600–900 W during operation and 3–4 kW during startup. In order to accommodate this pump and the other operating analyzers aboard the AMOG Surveyor, a 2.8 kW inverter is installed in the wheel well connected to two 100 Amp hr deep cycle – solar batteries that are capable of sourcing the startup surge power. All DC wiring utilizes 0/4A gauge cable, and the vehicle alternator is upgraded to provide 110 amps at idle and 220 amps (2.5 kW) at 1500 rpm. Except for the pump and 12V DC lighting and fan systems, all other power is routed through an uninterruptible power supply, with a regulated 12VDC 120 W power source for peripherals and other instruments to isolate from electrical noise sources

**Gas Handling**

As noted above, sample is pulled through the system using a scroll pump coupled to the analyzers using KF25 fittings to maximize gas conductance. Gas flow to the analyzers is controlled by adjustable throttle valves and bypass valves to optimize the sample measurement pressure, maintain a 5-10 Hz measurement rate and a short flow time from inlet to instrument. Sample is routed from a flexible mast atop the Surveyor to the analyzers using ½" stainless steel and Teflon® tubing. The latter is heated to 60°C to prevent condensation and allow for rapid sampling of gases that readily absorb onto surfaces (e.g., $NH_3$). The sampling height ranges from 3 to 5 m above ground level when the vehicle is moving fast or is stationary, respectively. In order to prevent the gas handling and analytical systems from fouling, a course filter is mounted at the sampling tube inlet, while a finer, 1 μm filter is mounted before splitting to the various analyzers.

**Gas Analysis**

As described in detail above, the ambient air is quantified using an Off-Axis ICOS Greenhouse Gas Analyzer that can provide measurements of $CH_4$, $CO_2$, and $H_2O$ at up to 10 Hz with 1σ, 5s precisions of 1 ppb for $CH_4$, 0.15 ppm for $CO_2$, and 100 ppm for $H_2O$ — the latter being useful for mapping airmass shifts. Additional cavity-enhanced analyzers also are simultaneously utilized in the AMOG Surveyor, including $NH_3$ and $NO_2$ analyzers (Los Gatos Research). The Surveyor has space, power, and gas handling support for a fourth and potentially fifth instrument as well.

**System Thermal and Noise Regulation**

In order to address the significant heat load generated by the scroll pump, an insulated compartment separates the pump from the other AMOG subsystems. This pump, inverter/battery compartment, and FGGA compartments are ventilated by 250, 90, and

CFM of forced air, respectively. Thermocouples continuously monitor the temperatures of the pump, inverter, analyzers, and vehicle compartments, and these readings are logged to confirm proper operation of cooling systems. For hot weather sampling missions, an auxiliary trunk air conditioner is incorporated, while windows are tinted.

Thermal insulation is coupled with noise insulation and reduction (e.g., pump mounts) that underlay vehicle choice. Specifically, an open vehicle (e.g., SUV or van) exposes the driver to significant noise, and as a result, the system may not be used as often. Current instrument cabin noise levels are comparable to road tire noise at 25 mph, highly tolerable, and thus facilitate continuous data collection.

**Peripheral Measurements**

Methane concentrations coupled with GPS coordinates alone are insufficient to identify the location and magnitude of methane sources. It is critical to correlate this spatially-resolved data with wind direction and speed. Thus, the AMOG Surveyor is equipped with a sonic anemometer that measures wind direction and speed with an accuracy of better than 2° and 0.1 m/s (or 1% of combined wind and vehicle velocity). To avoid vehicle flow streamline contamination of wind data, the anemometer is mounted 1.3 m above the vehicle roof (~3 m above ground) on a roof rack on a 1" diameter, stainless steel tube, which is three-way braced to reduce vibrations. The anemometer is surrounded by a wire cage, which does not affect wind measurements and provides protection against small branches and leaves. The AMOG Surveyor also includes a fiber-coupled solar spectrometer to support airborne remote sensing data over the UV-NIR range of 180-1080 nm.

**Data Communication**

In order to provide real-time mapping of the measured results, the AMOG Surveyor must be in constantly connected to the internet to download satellite imagery and maps; however, Google Earth allows up to 2 gigabytes of cached data. Thus, the Surveyor includes an amplified cellular modem to assure connectivity. Internal data communications are managed via high speed Ethernet connections and RS-232 serial servers. A separate data acquisition

[Figure]

*Figure 3. AMOG surveyor 2 Hz data during the discovery of a pipeline leak in an orchard while driving at highway speed on US 101; methane, $CH_4$, and ammonia, $NH_3$, are shown as color bars that are clamped to limits indicated in the legend. The altitude of the data point is also proportional to $CH_4$. Wind bar color and length scale with speed.*

system monitors the battery voltages, current flows, and GPS output at 1 Hz. Finally, the Surveyor is equipped with four, high-definition video cameras that record continuously and provide fore, aft, and side views from the vehicle. These video data can be correlated to specific measurements to help interpret methane sources, as well as enhancing vehicle security.

**Visualization and Data Logging Software**

Realtime data visualization is key to enabling successful adaptive surveying. Visualization of the spatial relationships between gas concentrations, winds, and other key parameters must be easily comprehensible to enable rapid and effective survey decisions. AMOG software pushes key data to Google Earth at 0.25 Hz to enable real-time visualization of multiple measurement components centered on the vehicle with automatic camera view settings. Older data is represented more transparently to facilitate clarity when doubling back. Near publication quality data visualizations are archived continuously, simplifying data reporting.

Flexibility and robustness in data logging are accomplished by custom software that communicates with all instruments aboard the Surveyor, using an asynchronous logging system based on a NMEA tagging protocol. This system provides robust data collection, even in cases where some signals are periodically lost (e.g., GPS) or mis-formatted. Serial buffers are frequently "flushed" (1 Hz or faster) to ensure that data recording times are well-defined: better than 0.1 s or equivalent to the best GPS accuracy of ~3 m while collecting data at fast highway speeds (30 m/s). Post-processing routines interpolate data to the FGGA data acquisition rate of 5 or 10 Hz, as well as identify and interpolate obvious outliers, and use more advanced digital filters to improve spatial and temporal resolution and suppress noise. In the case of lost temporary communication with instruments, internally logged data is merged with the tag stream in post processing to generate a seamless data product.

**Sample Data**

An example of a pipeline leak that was detected during a highway speed (25 m/s:~56mph) survey is shown in Figure 3, which was collected at 04:35 on the 10th of April 2014 UTC. While driving north on US 101, typical background methane levels suddenly rose to a peak of 8,900 ppb with the realtime wind vectors indicating a source to the north. Winds were light, ~0.5 m/s (~1.1mph) and shifted from offshore to onshore shortly before the plume as the highway crossed into a valley area, highlighting a recirculation pattern where the methane plume was re-transported back onshore.

Based on Google Earth imagery, a potential turnoff was located ~1/2 a mile further, allowing AMOG to slow safely and exit the highway to investigate further. Driving along an access road, methane levels of up to 21,000 ppb were re-encountered in an orchard close to a sign indicating a buried pipeline. Ammonia levels rose slowly rather than sharply—likely due to fertilizer. $CO_2$ levels (not shown) suggested some $CO_2$ in the natural gas leak, while $NO_2$

data confirmed the leak was unrelated to combustion. The post-processed data shown in Figure 3 are for 2 Hz. In contrast, realtime data is 0.25 Hz, which lessens visualization clutter, particularly for visualizations with multiple gases.

**Future Work**

Mobile monitoring shows great promise for identifying and containing methane leakage and emissions for both climate change mitigation and minimization of product loss. Future efforts will include making more compact and lower power mobile analysis systems for applications where speed is less critical, while measuring a wider array of components. One very promising concept involves simultaneous methane, ethane, and ammonia measurements to distinguish or even detangle methane emissions from pipeline leakage, landfills, diaries, and other sources that each contains specific ethane-to-methane and other gas ratios. Similarly, adding measurements of other greenhouse gases (e.g., carbon dioxide and nitrous oxide) and pollutants (e.g., carbon monoxide and nitrogen dioxide) can help elucidate climate variables and distinct pollution sources, while discriminating against road vehicular emission biases. Finally, multiple mobile monitors may be synchronized to provide wide-scale, real-time mapping of gas sources and leaks yielding far more accurate source strength and location data. **G&I**

[Figure]

**MANISH GUPTA** IS THE CHIEF TECHNOLOGY OFFICER AT LOS GATOS RESEARCH. MANISH HAS OVER 20 YEARS OF EXPERIENCE IN LASER SPECTROSCOPY AND ITS APPLICATION TO INDUSTRIAL, ENVIRONMENTAL, MEDICAL, AND MILITARY MONITORING. HE HOLDS A PH.D. IN PHYSICAL CHEMISTRY FROM HARVARD UNIVERSITY AND CAN BE CONTACTED AT M.GUPTA@LGRINC.COM.

[Figure]

**BRIAN LEEN, PH.D.,** IS A PRINCIPLE SCIENTIST AT LOS GATOS RESEARCH. BRIAN HAS BEEN ACTIVE IN THE DEVELOPMENT OF CAVITY ENHANCED TECHNOLOGIES AND THEIR APPLICATION TO MOBILE MONITORING FOR THE LAST 10 YEARS. HE HOLDS A PH.D. IN APPLIED PHYSICS FROM STANFORD UNIVERSITY.